# Enteroendocrine cell types that drive food reward and aversion

Ling Bai[1,2], Nilla Sivakumar[3], Shenliang Yu[1,2], Sheyda Mesgarzadeh[1,2], Tom Ding[1,2], Truong Ly[1,2,4], Timothy V Corpuz[3], James CR Grove[1,2,4], Brooke C Jarvie[1,2], Zachary A Knight[1,2,3,4]*

[1]Department of Physiology, University of California, San Francisco, San Francisco, United States; [2]Kavli Institute for Fundamental Neuroscience, University of California, San Francisco, San Francisco, United States; [3]Howard Hughes Medical Institute, University of California, San Francisco, San Francisco, United States; [4]Neuroscience Graduate Program, University of California, San Francisco, San Francisco, United States

**Abstract** Animals must learn through experience which foods are nutritious and should be consumed, and which are toxic and should be avoided. Enteroendocrine cells (EECs) are the principal chemosensors in the GI tract, but investigation of their role in behavior has been limited by the difficulty of selectively targeting these cells in vivo. Here, we describe an intersectional genetic approach for manipulating EEC subtypes in behaving mice. We show that multiple EEC subtypes inhibit food intake but have different effects on learning. Conditioned flavor preference is driven by release of cholecystokinin whereas conditioned taste aversion is mediated by serotonin and substance P. These positive and negative valence signals are transmitted by vagal and spinal afferents, respectively. These findings establish a cellular basis for how chemosensing in the gut drives learning about food.

*For correspondence:
zachary.knight@ucsf.edu

Competing interest: The authors declare that no competing interests exist.

## Editor's evaluation

This study provides insight into the functional diversity of specialized cells in the gut using a combination of transcriptomics, genetics, and behavioral assessment. It demonstrates how select enteroendocrine cell types mediate food reward while others drive aversion.

## Introduction

Although the desire to eat is innate, our food choices are driven by learning. Thus, the foods you enjoy today are different from your preferences as a child, in part because you have learned to associate specific flavors with their post-ingestive effects (*Sclafani and Ackroff, 2012*; *Yamamoto and Ueji, 2011*). This learning process causes nutrient-rich foods to become more rewarding and toxic substances to become aversive and is critical for guiding animals to choose safe and nutritious food sources in the wild (*Myers and Sclafani, 2006*). It also contributes to the motivational pull of energy-dense foods in modern society (*Johnson et al., 1991*; *Kern et al., 1993*). For these reasons, it is critical to establish where in the body ingested nutrients and toxins are detected to drive learning, and whether these parallel processes of food reward and aversion involve dedicated cell types, signals, and pathways.

One of the first sites of post-ingestive chemosensing is the epithelium of the small intestine, which contains a family of specialized sensory cells known as enteroendocrine cells (EECs) (*Adriaenssens et al., 2018*). EECs exist as multiple subtypes and express receptors and transporters that allow them

to monitor the chemical contents of the intestinal lumen. In response to detection of chemical cues, EECs release an array of hormones that include CCK, GIP, GLP-1, PYY, SST, and 5-HT. These molecules can act on local sensory neurons in the lamina propria (*Bellono et al., 2017*; *Kaelberer et al., 2018*; *Latorre et al., 2016*; *Ye et al., 2021*) or on distant organs via the circulation to modulate digestion, metabolism and appetite (*Adriaenssens et al., 2018*; *Gribble and Reimann, 2016*; *Najjar et al., 2020*). Thus, EECs link the detection of chemicals in the gut to the feedback regulation of behavior and physiology.

Given their sensory capabilities, EECs are attractive candidates to drive learning about the post-ingestive effects of different foods. However, investigation of their role in behavior has been hindered by the lack of methods for selectively manipulating these cells in vivo. EECs are sparsely scattered throughout the intestinal epithelium (~1% of cells), undergo rapid turnover (~1 week), and cannot be targeted with viral injections (*Polyak et al., 2008*). While there are marker genes that distinguish EEC subtypes from each other, these genes are also broadly expressed in other tissues that regulate metabolism and appetite, including, critically, sensory neurons throughout the gut. Thus, we lack the capability to directly probe the key nutrient sensors in the GI tract, creating a fundamental gap in our ability to study gut-brain signaling.

Here, we describe a general strategy for genetic access to a spectrum of EEC subtypes in vivo. We show that this approach can be used to target actuators to EECs selectively relative to all other cells in the body and thereby control the release of their natural cocktail of hormones at their endogenous sites in the intestine. Using this technique, we show that multiple EEC subtypes inhibit food intake but generate opposing valence signals that create either learned aversion or preference for specific foods. These positive and negative valence signals are transmitted by parallel gut-brain pathways that involve distinct neurotransmitters and afferent sensory neurons. These findings establish a cellular basis for how chemosensing in the GI tract is used to drive learning about food and provide a methodological resource for the further investigation of EEC function in vivo.

## Results

We first profiled the molecular diversity of EECs in the proximal intestine using single-cell RNA sequencing (scRNA-Seq). Due to their sparsity, EECs are difficult to characterize comprehensively even when sequencing is performed on a large scale (*Grün et al., 2015*; *Haber et al., 2017*). We therefore enriched for EECs prior to sequencing by labeling all cells derived from the common EEC progenitor using *Neurog3-Cre R26^LSL-tdTomato^* mice and then sorting tdTomato+ cells from the small intestine (*Figure 1A* and *Figure 1—figure supplement 1A, B*). This lineage-tracing method captures EECs at any stage after differentiation and yielded 7001 cells (with 2,168 EECs) for analysis. We then integrated our scRNA-seq data with a complementary dataset (2337 cells with 1244 EECs) that used a pulse-trace reporter *Neurog3^Chrono^* to transiently label EECs at an early stage after differentiation (*Gehart et al., 2019*). These two datasets differentially enriched for cell types depending on the timing after differentiation (*Figure 1A, D and E*) and together provide more complete coverage of EEC subtypes than either individually.

Unsupervised clustering analysis revealed eight clusters of EECs (*Figure 1B, C and F*). We observed two well-segregated clusters of enterochromaffin cells that correspond to early and late-stage cells (EC1 and EC2). These subtypes can be distinguished by the expression of *Tph1/Tac1* or *Tph1/Sct* but do not express any other peptide hormone (*Gehart et al., 2019*). We also observed three EEC clusters that share high expression of the peptide hormone *Cck* and *Sct*. Two of these clusters correspond to the canonical N/L cells (*Cck/Gcg/Pyy*) and I cells (*CCK*), whereas the third uniquely co-expresses *Cck* and *Tph1* (cluster EC/EEC). This third population has not been reported in other recent single-cell sequencing studies but likely corresponds to CCK+/5-HT+ cells that have been described immunohistochemically (*Reynaud et al., 2016*). Finally, we observed well-segregated clusters of D cells (*Sst*), K cells (*Gip*), and X cells (*Ghrl*) that are consistent with previous reports (*Gehart et al., 2019*). Thus, many canonical EEC subtypes can be differentiated from each other by expression of individual hormones (e.g. EC, D, K, and X cells), whereas others (N/L, I, EC/EEC cells) show partially overlapping hormonal expression.

### Expression of receptors and transporters in EEC subtypes

Different EEC subtypes displayed unique patterns of chemoreceptor and transporter expression (*Figure 2A*). For example, the receptors *Trpa1* and *Olfr558*, which detect irritants and microbial

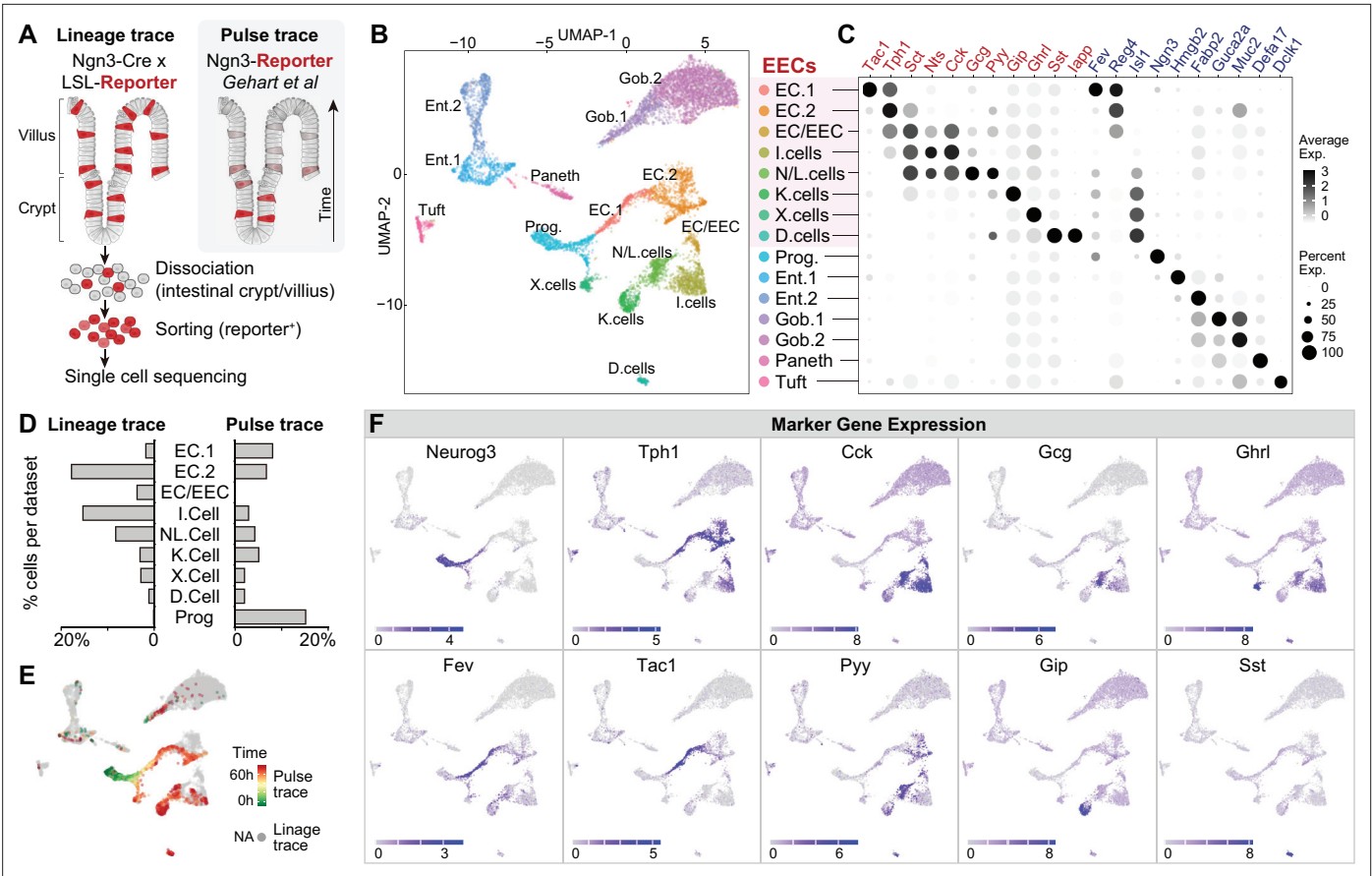

**Figure 1.** Molecular diversity of intestinal EECs. (**A**) Overview of strategies to enrich EECs for the scRNA-Seq. Lineage-trace or pulse-trace strategies permanently or transiently label EECs that are derived from *Neurog3* + progenitors. (**B**) Uniform manifold approximation and projection (UMAP) plots indicating cell subtype diversity across 9338 intestinal epithelium cells. This includes eight clusters of EEC subtypes (3412 cells, with 2168 lineage-traced cells and 1244 pulse-traced cells), progenitors, enterocytes, goblet cells, tuft cells, and Paneth cells. (**C**) Dot plot of expression of gut hormones (red) and cell type marker genes (blue) across different cell types in the intestinal epithelium. (**D**) The percentage of cells in each data set (lineage-trace and pulse-trace) that correspond to each EEC subtype. The lineage-trace strategy has comparatively fewer early stage cells (progenitors), whereas pulse-trace strategy did not label the EC-EEC cluster. (**E**) Differentiation time (from pulse-trace dataset) color-coded onto the UMAP map. (**F**) Expression of EEC cluster marker genes. UMAP plots show cells colored by gene expression using ln(TPM + 1), TPM: transcripts per million. See also *Figure 1—figure supplement 1* and *Figure 1—source data 1*.

The online version of this article includes the following source data and figure supplement(s) for figure 1:

**Source data 1.** Raw data for *Figure 1d*.

**Figure supplement 1.** Molecular diversity of intestinal EECs.

metabolites, respectively, were selectively expressed in enterochromaffin cells (*Bellono et al., 2017*), whereas nutrient-sensing I/K/N/L cells expressed an array of receptors and transporters involved in the detection of sugars, fats, and amino acids (*Figure 2A*). Of note, CCK+ EECs were recently reported to release glutamate at synapses with sensory afferents, thereby enabling the rapid control of feeding behavior (*Buchanan et al., 2022*; *Kaelberer et al., 2018*). However, we were unable to detect the expression of vesicular glutamate transporters (*Slc17a6*, *Slc17a7*, and *Slc17a8*) in any cell type within the intestinal epithelium by scRNA-Seq (*Figure 2A*).

To improve our ability to detect low abundance genes in CCK+ EECs, we crossed *Cck^{Cre}* mice to a *R26^{LSL-tdTomato}* reporter, purified tdTomato+ and tdTomato- cells from the small intestine epithelium by flow cytometry, and then performed RNA-Seq on the pooled, sorted cells (*Figure 2B–D*). As expected, many EEC marker genes were highly enriched in the tdTomato+ cells, confirming the efficacy of the cell sorting (e.g. *Cck*, 440-fold enriched in tdTomato+ versus tdTomato- cells; *Figure 2C* and *Figure 2—figure supplement 1*). We also detected numerous receptors and transporters previously reported to be important for nutrient sensing in EECs (*Figure 2E* and *Figure 2—figure*

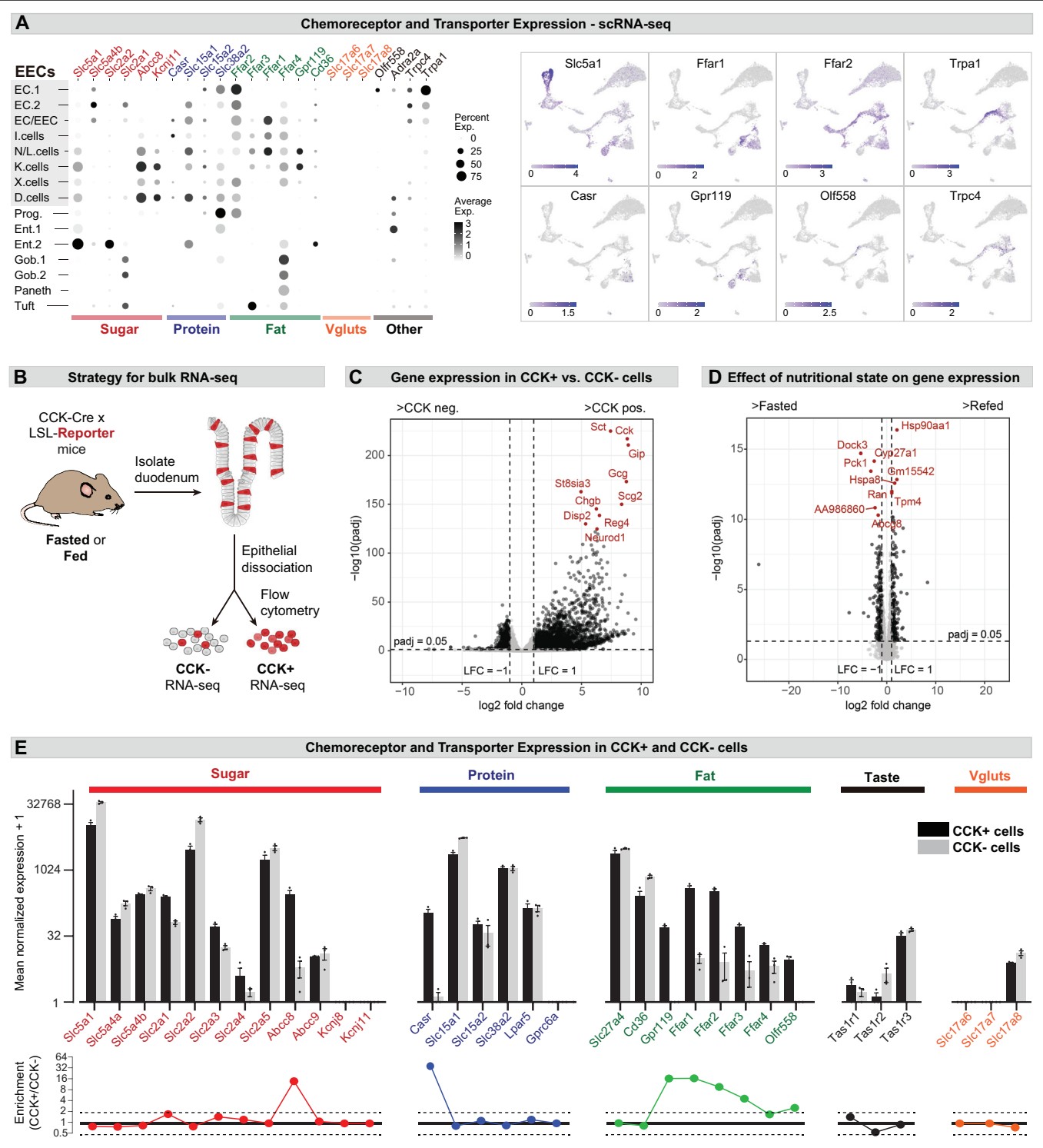

**Figure 2.** Expression of chemoreceptors and transporters in EECs. (**A**) Expression of genes involved in the detection or absorption of nutrients and other chemical signals, as detected in scRNA-Seq experiments described in *Figure 1*. Left: dot plot shows gene expression across clusters. Right: UMAP plots show cells colored by gene expression using ln(TPM + 1). (**B**) Strategy for isolation of cells from duodenal epithelium labeled by *CCK^Cre* (tdTomato+) or not (tdTomato-) for pooled RNA-Seq. (**C**) Volcano plot showing the log2 fold change (LFC) and adjusted p-values (padj) of differentially expressed genes between CCK+ (tdTomato+) and CCK- (tdTomato-) cells. Top 10 most significant genes are highlighted in red and include many EEC markers. (**D**) Volcano plot showing the log2 fold change (LFC) and adjusted p-values (padj) of differentially expressed genes between refed and fasted conditions in CCK+ cells. Top 10 most significant genes are highlighted in red. Few genes involved in nutrient transport or sensing were identified as highly modulated by fasting. (**E**) Normalized for genes involved the detection or transport of sugar, protein and fat, as well as taste receptor genes

*Figure 2 continued on next page*

*Figure 2 continued*

and glutamate transporters. Note the log2 scale of the y-axis. Expression in CCK+ cells (tdTomato+) and CCK- cells (tdTomato-) from fed mice shown in black and gray. Bottom: Ratio of expression of each gene in CCK+ and CCK- cells. Values greater than 1 indicate enrichment in CCK+ cells. Values reported as mean ± SEM. See also *Figure 2—figure supplement 1* and *Figure 2—figure supplement 2*.

The online version of this article includes the following figure supplement(s) for figure 2:

**Figure supplement 1.** Gene expression in CCK+ EECs.

**Figure supplement 2.** Expression of vesicular glutamate transporters in the small intestine.

*supplement 1*). The expression level of these chemoreceptors spanned more than five orders of magnitude and included genes selectively expressed in CCK+ cells (such as the amino acid sensor *Casr*, which was 620-fold enriched in tdTomato+ versus tdTomato- cells) as well as other genes more broadly expressed throughout the intestinal epithelium (such as *Slc5a1*, which encodes the glucose transporter SGLT1). We performed these experiments in both fasted and fed mice, but nutritional state had little effect on the expression of most genes involved in nutrient sensing or signaling in EECs (*Figure 2D* and *Figure 2—figure supplement 1*).

The glutamate transporter *Slc17a8* (VGLUT3) was detected at a low level in both tdTomato+ and tdTomato- cells, whereas mRNAs for *Slc17a6* (VGLUT2) and *Slc17a7* (VGLUT1) were not detected in any sample (*Figure 2E*). To further characterize the expression of these genes, we crossed Cre drivers for each glutamate transporter (*Slc17a6^Cre*, *Slc17a7^Cre*, and *Slc17a8^Cre*) to a *R26^LSL-tdTomato* reporter and then imaged sections from the proximal intestine. *Slc17a7^Cre* recombination labeled primarily axons innervating the intestinal villi, consistent with VGLUT1 expression in sensory neurons. For *Slc17a6^Cre* and *Slc17a8^Cre*, we observed some labeled cells within the villi, but these were localized primarily to the lamina propria rather than the epithelium (*Figure 2—figure supplement 2*). Consistently, these tdTomato+ cells appeared spherical and lacked the elongated, 'flask-shaped' morphology characteristic of open-type EECs (*Liddle, 2018*). They also showed no overlap with 5-HT, which labels all enterochromaffin cells and ~40% of CCK+ cells (*Figure 2—figure supplement 2*). This suggests that glutamate release by EECs may be restricted to rare cells or involve novel, VGLUT-independent mechanisms for loading glutamate into synaptic vesicles.

## An intersectional genetic approach to selectively target EEC subtypes

While EEC subtypes can be distinguished from other epithelial cells via marker genes (*Figure 1C and F*), the Cre lines corresponding to those genes also label other cell types throughout the body (*Figure 3—figure supplement 1*). For example, *Cck^Cre* and *Tac1^Cre*, which demarcate major subclasses of EECs (discussed below), also label sensory and motor neurons that innervate the intestinal villi (*Figure 3D–E*). For *Cck^Cre*, this labeling is due to widespread recombination in the enteric nervous system (*Figure 3—figure supplement 1C* and *Morarach et al., 2021*) whereas *Tac1^Cre* labels subsets of spinal, vagal, and enteric neurons (*Browne et al., 2017*; *Kupari et al., 2019*; *Morarach et al., 2021*). This recombination is so widespread that even the focal delivery of light to the intestine would be insufficient to selectively target these EECs for functional manipulations in vivo.

To address this, we developed an intersectional genetic approach for targeting EEC subtypes selectively relative to all other cells in the body (*Figure 3A*). We first generated a *Villin-Flp* transgenic mouse that efficiently labels the intestinal epithelium (>99%). We extensively characterized the specificity of this line to show that it does not label other tissues, including non-epithelial cells, the nervous system, and visceral organs (*Figure 3B, C*), with the exception of rare cells in pancreas (~1%). We then crossed this *Villin-Flp* mouse to Cre drivers that target distinct subsets of cells within the epithelium (described below), such that Flp and Cre act as sequential selectivity filters that together label individual EEC subtypes (*Figure 3A, F and G*). Importantly, we confirmed for each Cre driver that it elicits no overlapping recombination in pancreas when combined with *Villin-Flp* (*Figure 3—figure supplement 2*). Thus, these Cre and Flp drivers can be used in conjunction with dual-recombinase reporter mice to deliver actuators to dispersed EEC subtypes that express both Cre and Flp.

We next assembled a panel of Cre driver mice corresponding to genes expressed in different subsets of EECs, or their progenitors, and characterized their recombination pattern in the intestinal epithelium. Among EEC progenitor markers, we found that *Fev-Cre* efficiently labels all EEC subtypes tested (~90% of cells) but not other cell types in the intestinal epithelium (*Figure 3H, I*, and

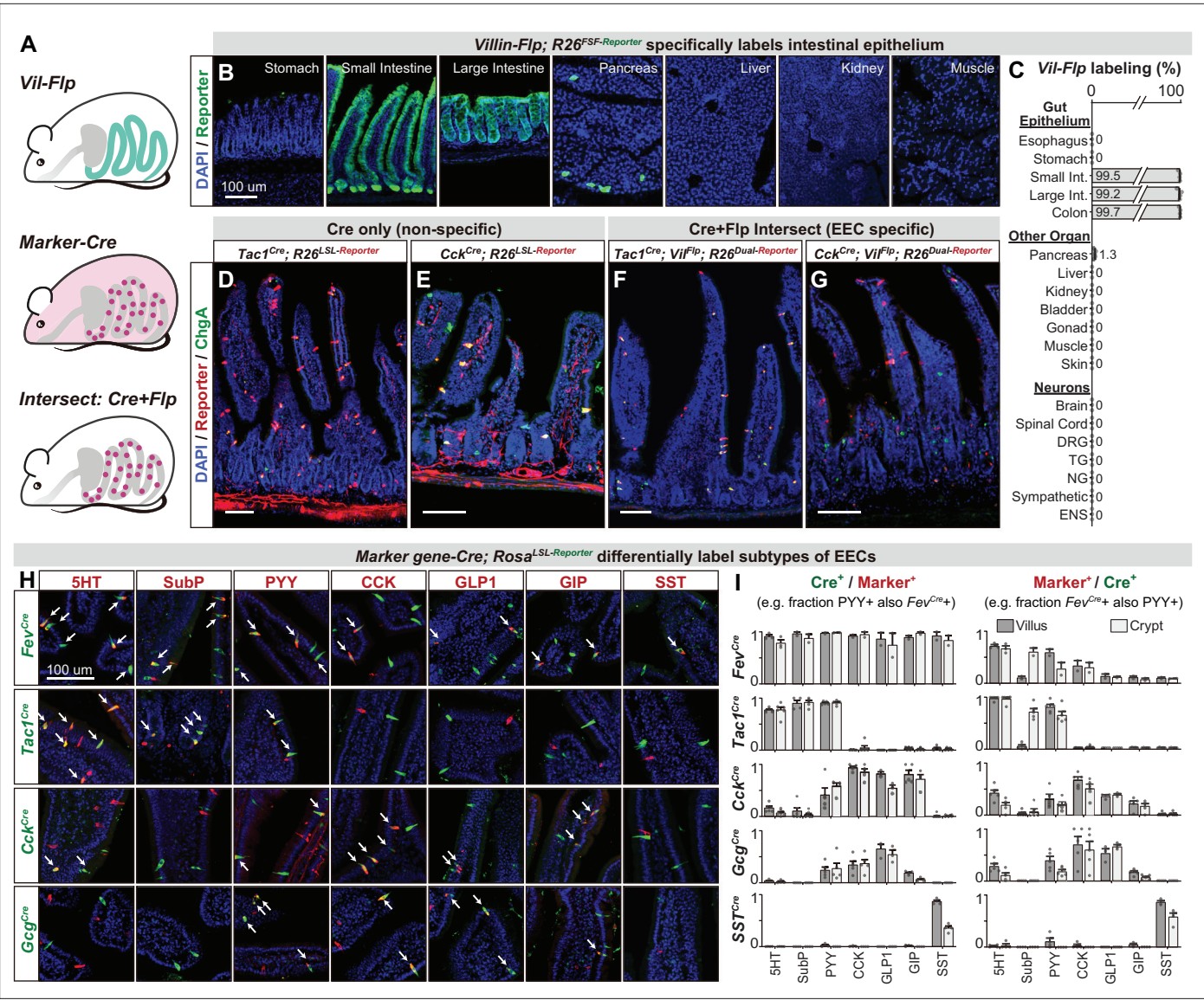

**Figure 3.** An intersectional genetic approach to selectively target EEC subtypes. (**A**) Schematic illustrating the intersectional genetic approach to target EECs. *Vil-Flp* labels the intestinal epithelium whereas *Marker gene-Cre* labels diverse cells throughout the body. Their intersection defines EECs. (**B**) Cross-section of organs showing that *Vil-Flp* promotes recombination and reporter expression in the intestinal epithelium, as well as very small subsets of pancreatic cells, but not liver, kidney, muscle, or stomach. (**C**) Quantification of the Vil-Flp labeling across the GI tract, other visceral organs, and neuronal tissues reveals high selectivity for the intestine. (**D–G**) Cross-sections of the small intestines reveal the recombination pattern of single *Marker gene-Cre* lines (**D–E**) or the *Marker gene-Cre; Vil-Flp* intersectional strategy (**F–G**). Note that *Tac1Cre* (**D**) and *CckCre* (**E**) label not only EECs but also other cell types, including neuronal terminals and cell bodies in the intestine. In contrast, the labeling using the intersectional strategy is restricted to EECs (**F–G**). ChgA staining (green) marks the majority of enterochromaffin cells and subsets of EECs. (**H**) Double immunostaining of hormonal marker genes (red) and Cre-induced reporter (green) in small intestine villus sections of various *Marker gene-Cre; R26LSL-Reporter* mice. White arrows indicate colocalization. (**I**) Quantification of co-localization between different EEC subtypes and the cells labeled by various *Marker gene-Cre; R26LSL-Reporter* mice (example images are shown in 3H). The bar graphs on the left show the fraction of EEC subtypes (Marker+) that are labeled by each Cre line. For example, the fraction of cells that stained for PYY that are also labeled by *Fev-Cre* recombination. The bar graphs on the right show the fraction of the cells labeled by each Cre line that are also stained for the marker. For example, the fraction of cells labeled by *Fev-Cre* recombination that also stained for PYY. Comparing the left and right reveals, for example, that *Fev-Cre* labels a high percentage of cells for all EEC subtypes, but that each EEC subtype alone comprises only a fraction of *Fev-Cre* labeled cells. Each quantification is performed separately for cells in the villi (dark gray) and crypts (light gray). Values are reported as mean ± SEM. Scale bar: 100 μm. See also *Figure 3—figure supplements 1 and 2*, and *Figure 3—source data 1*.

The online version of this article includes the following source data and figure supplement(s) for figure 3:

**Source data 1.** Raw data for *Figure 3I*.

**Figure supplement 1.** Summary of the labeling patterns of Marker gene-Cre lines.

*Figure 3 continued on next page*

*Figure 3 continued*

**Figure supplement 2.** An intersectional genetic approach to selectively target EEC subtypes.

**Figure supplement 2—source data 1.** Raw data.

*Figure 3—figure supplement 2*). Thus, this line enables targeting all EECs simultaneously. In contrast, the progenitor markers *Neurog3-Cre* and *Isl1^Cre^* labeled other epithelial cells (*Figure 1B*) or the stem cell niche (*Figure 3—figure supplement 2*) and therefore lacked useful specificity.

We also tested Cre drivers corresponding to EEC peptide hormones, which revealed that *Tac1^Cre^*, *Cck^Cre^*, and *Sst^Cre^* label distinct EEC subsets (*Figure 3H, I*, and *Figure 3—figure supplement 2A*). *Tac1^Cre^* specifically and efficiently labels 5-HT producing enterochromaffin cells, including both early stage (TAC1+, cluster EC.1) and late stage (which have lost TAC1 expression, cluster EC.2) cells but not EECs that express the peptides CCK, GLP-1, GIP, or SST. On the other hand, *Cck^Cre^* labels virtually all CCK+ cells (~98% in villi) and majority of GLP-1 and GIP expressing cells, consistent with our scRNA-seq data (*Figure 1F*). Interestingly, we found that subsets of *Cck^Cre^* labeled EECs also co-express 5-HT (*Figure 2H*). These cells are distinct from the 5-HT cells derived from the TAC1 + lineage and likely correspond to cluster EC/EEC in our scRNA-seq data (*Figure 1B and C*). A similar recombination pattern to *Cck^Cre^* was observed for *Gcg^Cre^*. Finally, *Sst^Cre^* specifically targets D cells (SST+) without any labeling of other cell types. Thus, these Cre drivers can be used in conjunction with *Villin-Flp* to provide efficient and specific genetic access to a spectrum of different EEC subsets.

## Distinct EEC subtypes signal reward and aversion

We applied this intersectional strategy to manipulate EEC subtypes and investigate how they influence feeding behavior. As hormone release from EECs is naturally triggered by Gq-signaling (*Adriaenssens et al., 2018*), we targeted the DREADD hM3Dq to EECs (by crossing Cre drivers to *Villin-Flp* and *R26^Dual-hM3Dq^* mice) and stimulated these cells with CNO (*Figure 4A*). We found that CNO activation of all EECs (labeled by *Fev-Cre*) caused a robust reduction of food intake in fasted mice, whereas CNO alone had no effect in littermate controls (*Figure 4B and C*). We further showed that several EEC subtypes were sufficient to inhibit food intake when activated individually, including *Tac1^Cre^* cells (enterochromaffin cells) as well as *Cck^Cre^* and *Gcg^Cre^* cells (I/NL/K cells) (*Figure 4B, C*). In contrast, there was no effect on food intake following stimulation of *Sst^Cre^* cells (D cells; *Figure 4B, C*) or X cells that express ghrelin (via *Ghrl^Cre^*; *R26^LSL-hM3D^* mice; *Figure 4B-D*). The lack of response in X cells may reflect the fact that ghrelin stimulates feeding only when administered at supraphysiologic levels (~20-fold above baseline, *McFarlane et al., 2014*).

We next investigated the ability of EEC subtypes to drive learning about food. We first asked whether pairing EEC activation with consumption of a strongly preferred flavor could create a conditioned taste aversion (CTA). Animals were trained on consecutive days by injection of CNO after sucrose ingestion or vehicle after water ingestion, and then tested for their learned preference (*Figure 4E*). We found that activation of all intestinal EECs labeled by *Fev-Cre* induced dramatic aversion, such that animals avoided consumption of sucrose after just one training session (*Figure 4F*) and completely reversed their preference in a subsequent two-bottle test (*Figure 4G*). This learning was absent in littermate controls (lacking hM3D expression) treated with CNO (*Figure 4F-G*). Testing of individual subtypes revealed that activation of *Tac1^Cre^*-labeled enterochromaffin cells alone was sufficient to drive this CTA, whereas activation of other subtypes (labeled by *Cck^Cre^*, *Gcg^Cre^*, or *Sst^Cre^*) created no measurable aversion (*Figure 4F-G*).

To test whether activation of these other EEC subtypes may encode positive valence, we next used an assay for conditioned flavor preference (CFP). Mice were trained by allowing them to consume two neutral, non-nutritive flavors, one of which contained CNO and the other vehicle, and learning was accessed by comparing their flavor preference before and after training (*Figure 4H*). This revealed that activation of EECs labeled by *Cck^Cre^* or *Gcg^Cre^* was sufficient to induce robust conditioned preference (*Figure 4I*), whereas activation of D cells labeled by *Sst^Cre^* had no effect (*Figure 4I*). *Fev-Cre* or *Tac1^Cre^* were not re-tested in this CFP assay, because we had already shown they promote dramatic aversion in the more stringent CTA test (*Figure 4F and G*). Together, these results reveal that multiple EEC subtypes inhibit food intake but generate opposing valence signals that create learned aversion or preference for specific foods (*Figure 4J*).

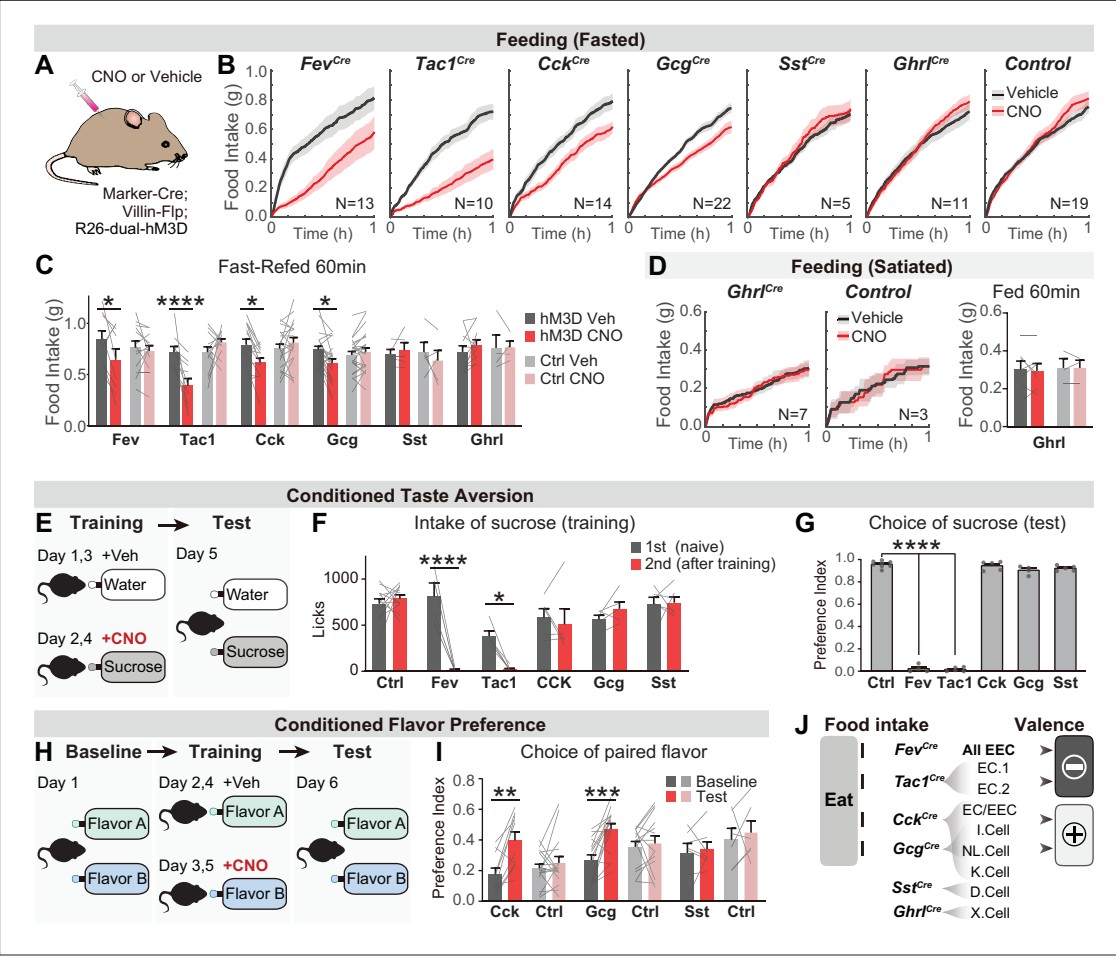

**Figure 4.** Distinct EEC subtypes signal reward and aversion. (**A**) Overview of the chemogenetic approach to stimulate EEC subtypes. Littermates that were Cre or Flp negative (and therefore did not express hM3D) were used as genetic controls. (**B**) Cumulative food intake over one hour of overnight food-deprived mice, comparing trials with CNO or saline treatment. (**C**) Quantification of total food intake at 1 hr from (**B**), including data for both triple-mutants and littermate controls. Two-way repeated measures ANOVA. Genotype: $F_{(11, 138)}=1.508$, $p=0.135$. Drug: $F_{(1, 138)}=8.392$, $p=0.0044$. Interaction: $F_{(11, 138)}=5.5157$, $p<0.0001$. (**D**) Cumulative food intake of satiated mice, and quantification of total food intake at 1 hr. (**E**) CTA paradigm. In the training trials, CNO or vehicle is delivered intraperitoneally after 20 min of sucrose or water ingestion. This is followed by a two-bottle test on day 5. (**F**) Total sucrose intake during training trials of CTA, comparing the 1st trial (day 1, naive mice) versus the 2nd trial (day 3, after pairing sucrose with CNO for one trial). Two-way repeated measures ANOVA. Day: $F_{(1, 63)}=16.31$, $p=0.0001$. Genotype: $F_{(5, 63)}=3.463$, $p=0.0079$. Interaction: $F_{(5, 63)}=3.382$, $p=0.0091$. (**G**) Preference for sucrose during the two-bottle test (day 5). Ordinary one-way ANOVA. $F_{(5, 25)}=1168$, $p<0.0001$. (**H**) CFP paradigm. In the training trials, CNO or vehicle is delivered in the drinking solution along with flavor A or flavor B. (**I**) Preference for the CNO paired in a two-bottle test at baseline (day 1) and after CFP training (day 6). Two-way repeated measures ANOVA. Genotype: $F_{(5, 63)}=3.463$, $p=0.0079$. Day: $F_{(1, 63)}=16.31$, $p=0.0001$. Interaction: $F_{(5, 63)}=3.382$, $p=0.0091$. (**J**) Summary of EEC subtypes labeled by each Cre line, and their effects on food intake and food learning. Values are reported as mean ± SEM. N mice is annotated within figures. *$p<0.05$, **$p<0.01$, ***$p<0.001$, ****$p<0.0001$, two-way ANOVA (C, D, G, and I) or Ordinary one-way ANOVA (**F**). See also *Source data 1* and *Figure 4—source data 1*, *Figure 4—source data 2*, *Figure 5—source data 3*, *Figure 4—source data 4*.

The online version of this article includes the following source data for figure 4:

**Source data 1.** Raw data for *Figure 4C*.

**Source data 2.** Raw data for *Figure 4F*.

**Source data 3.** Raw data for *Figure 4G*.

**Source data 4.** Raw data for *Figure 4I*.

## Different neurotransmitters regulate food consumption and learning

We next investigated the gut hormones that mediate these effects, by combining chemogenetic activation of EEC subtypes with administration of receptor antagonists. *Tac1Cre* labels enterochromaffin cells that release 5-HT, substance P (encoded by the *Tac1* gene), and PYY, while *CckCre* labels EECs

that release CCK, GLP-1, GIP, PYY, and 5-HT (*Figure 3H, I*, and *Figure 3—figure supplement 1A*). We therefore tested a panel of antagonists targeting key receptors for these hormones, including ondansetron (for 5-HT3R), RP67580 (for TACR1), and JNJ-31020028 (for the PYY receptor NPY2R), Devazepide (for CCKAR) and Ex3 (for GLP1R).

We found that the acute inhibition of feeding caused by enterochromaffin cell stimulation (*Tac1^Cre*) was attenuated by treatment with an antagonist targeting 5-HT3R but not TACR1 or NPY2R (*Figure 5A* and *Figure 5—figure supplement 1A*). Interestingly, although 40% of *Cck^Cre* EECs also secrete 5-HT (*Figure 3I*), a 5-HT3R antagonist had no effect on the inhibition of feeding caused by *Cck^Cre* EECs. In contrast, this feeding inhibition was fully blocked by antagonists against either CCKAR or NPY2R, but not TACR1 or GLP1R (*Figure 5B* and *Figure 5—figure supplement 1B*). Thus, although the hormones released by *Tac1^Cre* EECs and *Cck^Cre* EECs are partially overlapping, the receptors required for their acute inhibition of feeding are different (5-HT3R for *Tac1^Cre* EECs, versus CCKAR and NPY2R for *Cck^Cre* EECs) (*Figure 5E*). This suggests that the same gut hormone can have different effects when released by different cell types.

To investigate how these signals drive learning, we treated animals with the same panel of antagonists during taste conditioning. We found that inhibition of 5-HT3R or TACR1 during training could partially block the CTA induced by enterochromaffin cell activation, whereas the NPY2R antagonist had no effect (*Figure 5C* and *Figure 5—figure supplement 1E*). Strikingly, combined treatment with 5-HT3R and TACR1 antagonists during training abolished the enterochromaffin cell induced CTA (*Figure 5C* and *Figure 5—figure supplement 1E*). Similar effects were observed for the CTA induced by activation of all EECs (*Figure 5—figure supplement 1C, D*). Thus, enterochromaffin cells can induce learned aversion by signaling through the 5-HT3R and TAC1R, whereas their acute inhibition of feeding requires only 5-HT/5-HT3R signals (*Figure 5E*).

We performed a similar analysis for the effects of *Cck^Cre* cells. Since CCKAR and NPY2R are required for the feeding inhibition induced by *Cck^Cre* cells, we tested whether these receptors are necessary for flavor conditioning by administering antagonists during training. We found that flavor preference induced by *Cck^Cre* cell stimulation was prevented by blocking CCKAR but not NPY2R (*Figure 5D*). The fact that NPY2R antagonists can block *Cck^Cre*-induced satiety, but not flavor preference, reveals these cells influence satiety and learning through separable mechanisms (*Figure 5E*).

## Parallel neuronal pathways convey EEC signals to the brain

We sought to identify how this information is transmitted to the brain. Vagal and spinal afferents comprise the two major ascending circuits for gut-brain communication (*Brookes et al., 2013*; *Latorre et al., 2016*), and subsets of these sensory neurons express receptors for hormones released by EECs, including 5-HT3R, CCKAR, and NPY2R (*Bai et al., 2019*; *Li et al., 2016*; *Sharma et al., 2020*; *Williams et al., 2016*). We therefore tested the necessity of these two pathways for the effects of EECs on learning and appetite.

We prepared cohorts of triple transgenic mice for chemogenetic activation of *Cck^Cre* or *Tac1^Cre*-labeled cells, and then subjected these animals to one of two procedures (*Figure 6A*): subdiaphragmatic vagotomy to surgically remove vagal innervation to the abdominal viscera (*Figure 6—figure supplement 1A, B*) or intrathecal injection of resiniferatoxin (RTX), a potent TRPV1 agonist, to ablate TRPV1+ spinal afferent neurons without impacting vagal sensory neurons (*Figure 6—figure supplement 1C, D*; *Mishra and Hoon, 2010*). We then measured the effect of chemogenetic stimulation of EECs in these deafferented animals and controls.

We found that vagotomy abolished the feeding inhibition induced by activation of *Cck^Cre* labeled EECs (*Figure 6C* and *Figure 6—figure supplement 1F*) but had no effect on the reduction in feeding induced by *Tac1^Cre* EECs (*Figure 6B* and *Figure 6—figure supplement 1E*). In contrast, intrathecal RTX treatment abolished the feeding inhibition induced by the *Tac1^Cre* labeled EECs (*Figure 6D* and *Figure 6—figure supplement 1G*) but had no effect on responses to *Cck^Cre* EECs (*Figure 6E* and *Figure 6—figure supplement 1H*). Thus, the EECs labeled by *Tac1^Cre* and *Cck^Cre* inhibit food intake via spinal and vagal pathways, respectively (*Figure 6H*).

We next investigated how these two neuronal pathways contribute to EEC driven learning. Intrathecal RTX prior to training was sufficient to partially prevent the acquisition of CTA following *Tac1^Cre* activation, indicating that spinal afferents contribute to this process (*Figure 6F*), whereas vagotomy prior to training did not block this CTA (as reflected by the reduction in sucrose intake after one round

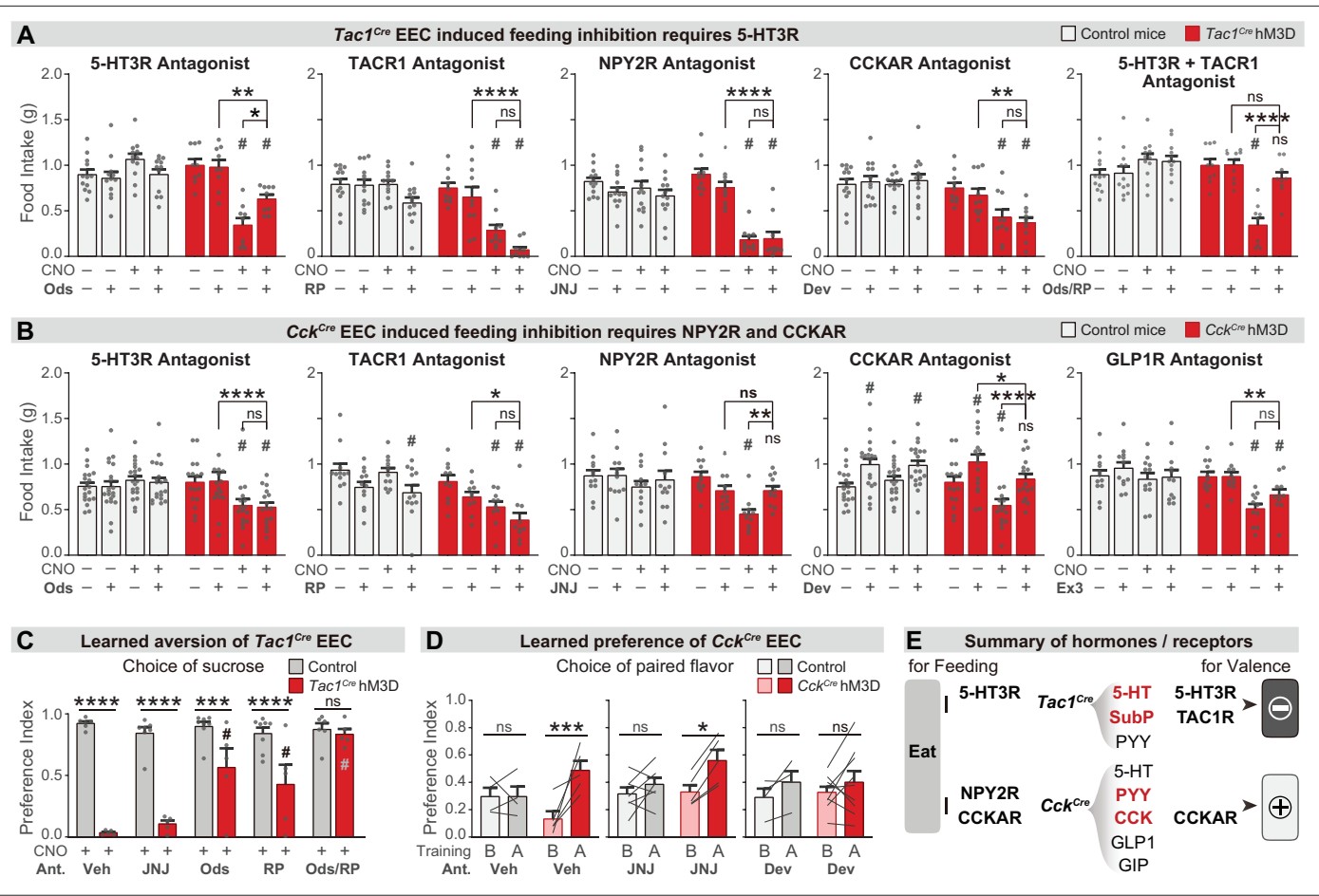

**Figure 5.** Different neurotransmitters regulate food consumption and learning. (A–B) One-hour food intake of overnight food-deprived mice treated with either CNO or vehicle, and the receptor antagonist or vehicle, immediately before food access. Experimental *Tac1^{Cre}; Vil-Flp; R26^{Dual-hM3D}* mice (A), *Cck^{Cre}; Vil-Flp; R26^{Dual-hM3D}* mice mice (B) are shown in red. Littermate control mice that do not express hM3D are shown in gray (A,B). The key comparison is the last two red bars in each set, which reports whether the receptor antagonist can block the ability of CNO to reduce food intake. Statistics for (A): Two-way repeated measures ANOVA. Ods. Genotype: F(1, 19)=14.25, p=0.0013. Drug: F(3, 57)=6.741, p=0.0006. Interaction: F(3, 57)=19.00, p<0.0001. RP. Genotype: F(1, 21)=34.88, p<0.0001. Drug: F(3, 63)=21.57, p<0.0001. Interaction: F(3, 63)=8.333, p<0.0001. JNJ. Genotype: F(1, 21)=21.84, p=0.0001. Drug: F(3, 63)=24.65, p<0.0001. Interaction: F(3, 63)=16.17, p<0.0001. Dev. Genotype: F(1, 21)=23.33, p<0.0001. Drug: F(3, 63)=4.295, p=0.0081. Interaction: F(3, 63)=5.032, p=0.0034. Ods RP. Genotype: F(1, 19)=8.406, p=0.0092. Drug: F(3, 57)=8.835, p<0.0001. Interaction: F(3, 57)=21.44, p<0.0001. Statistics for (B): Two-way repeated measures ANOVA. Ods. Genotype: F(1, 34)=2.679, p=0.1109. Drug: F(3, 102)=4.382, p=0.0061. Interaction: F(3, 102)=10.11, p<0.0001. RP. Genotype: F(1, 20)=14.75, p=0.0010. Drug: F(3, 60)=10.02, p<0.0001. Interaction: F(3, 60)=2.394, p=0.0772. JNJ. Genotype: F(1, 21)=4.676, p=0.0423. Drug: F(3, 63)=8.661, p<0.0001. F(3, 63)=2.446. Interaction: p=0.0720. Dev. Genotype: F(1, 34)=1.694, p=0.2018. Drug: F(3, 102)=24.64, p<0.0001. Interaction: F(3, 102)=7.190, p=0.0002. Ex. Genotype: F(1, 21)=4.407, p=0.0481. Drug: F(3, 63)=12.92, p<0.0001. Interaction: F(3, 63)=5.193, p=0.0029. See *Source data 1* for exact p values of all post-hoc tests. (C) *Tac1^{Cre}; Vil-Flp; R26^{Dual-hM3D}* mice (red) were subjected to CTA training as in *Figure 4*, except that they were treated with the indicated antagonist or vehicle prior to each training session. Littermate controls that lack hM3D expression (gray) were treated identically. Shown is the preference for sucrose during the two-bottle test on day 5. Note that all mice received CNO during CTA training. Two-way repeated measures ANOVA. Drug: F(4, 50)=10.62, p<0.0001. Genotype: F(1, 50)=117.6, p<0.0001. Interaction: F(4, 50)=11.15, p<0.0001. (D) *Cck^{Cre}; Vil-Flp; R26^{Dual-hM3D}* mice mice (red) were subjected to CFP training as in *Figure 4*, except that they were treated with the indicated antagonist or vehicle prior to each training session. Littermate controls that lack hM3D expression (gray) were treated identically. Results are shown for the two-bottle tests on day 1 (before training, 'B') and day 6 (after training, 'A'). Preference index indicates the preference for the CNO paired flavor, as in *Figure 4*. Note that all animals received CNO during training. Two-way repeated measures ANOVA. Time: F(1, 28)=22.71, p<0.0001. Genotype-Treatment: F(5, 28)=0.7891, p=0.5664. Interaction: F(5, 28)=2.895, p=0.0314. (E) Summary of gut hormones expressed in the *Tac1^{Cre}* EECs or *Cck^{Cre}* EECs, and the corresponding receptors required for the regulation of feeding behavior or learned preferences, based on the pharmacological data presented in this figure. Values are reported as mean ± SEM. *p<0.05, **p<0.01, ***p<0.001, ****p<0.0001; # or ns above bars indicate p-value comparing with the vehicle control treatment of the same genotype (#p<0.05); two-way ANOVA. See also *Figure 5—figure supplement 1*, figure , *Figure 5—source data 1*, *Figure 5—source data 2*, *Figure 5—source data 3*.

The online version of this article includes the following source data and figure supplement(s) for figure 5:

*Figure 5 continued on next page*

*Figure 5 continued*

**Source data 1.** Raw data for *Figure 5A*.

**Source data 2.** Raw data for *Figure 5B*.

**Source data 3.** Raw data for *Figure 5C*.

**Source data 4.** Raw data for *Figure 5D*.

**Figure supplement 1.** Different neurotransmitters regulate food consumption and learning.

**Figure supplement 1—source data 1.** Raw data.

of training; *Figure 6F* and *Figure 6—figure supplement 1I*). In contrast, the conditioned flavor preference induced by the *Cck^Cre* activation was unaffected by the ablation of TRPV1 +spinal afferents but abolished by vagotomy (*Figure 6G*). Thus, vagal afferents are necessary for the *Cck^Cre* EECs to transmit positive valence to the brain, whereas enterochromaffin-cell-mediated aversion is relayed at least in part by spinal pathways (*Figure 6H*).

## Discussion

EECs are the proximal sensors of most chemical cues in the GI tract and consequently play a privileged role in gut-brain communication. However, direct tests of their function have been limited by the difficulty of selectively accessing these cells for manipulations in vivo. One recent study used the gene *Insl5* to target GLP1/PYY+ EECs in the colon (*Lewis et al., 2020*), but most marker genes that label EEC subtypes are broadly expressed in other tissues important for metabolism and appetite (*Figure 3—figure supplement 1*). As a result, EECs have not been subject to the kinds of acute and targeted manipulations that have been critical for unravelling how circuits in the brain and peripheral nervous system control feeding behavior.

Here, we have described a general strategy for genetic access to a spectrum of EEC subtypes for functional manipulations in behaving mice. This approach uses Flp and Cre drivers as sequential selectivity filters to progressively restrict transgene expression to the intestinal epithelium and then to cell types within that structure. This makes it possible to target rare EEC subsets (each <1% of epithelial cells) with high efficiency (>90% labeling) and undetectable recombination in off-target tissues. Importantly, these tools can be combined with a variety of existing dual recombinase reporter mice to deliver actuators (e.g. DREADDs or opsins) to different EEC subsets, thereby opening the door to systematic functional exploration of these cells in vivo.

For this study, we assembled a panel of Cre drivers that label either all EECs (*Fev-Cre*) or nonoverlapping subsets that collectively span much of EEC diversity (*Cck^Cre*, *Tac1^Cre*, *Sst^Cre*, and *Ghrl^Cre*). While these markers define canonical EEC subtypes, other classes of Cre drivers could be used to investigate cross-sections of EECs that vary along different dimensions. For example, the expression of nutrient receptors and transporters may define EEC subpopulations that have a shared chemosensory function (*Figure 2A*). Similarly, in this study, we have not explored how the function of these molecularly defined EEC subsets varies with location in the intestine, but this could be addressed by combining the genetic approach described here with tools for wireless optogenetics in visceral organs (*Park et al., 2015*).

### Manipulations of hormones versus cell types

Our understanding of how EECs modulate behavior is based primarily on studies of the hormones they produce (*Adriaenssens et al., 2018*; *Clemmensen et al., 2017*; *Myers and Sclafani, 2006*). While much has been learned from this pioneering work, all EECs express more than one hormone (*Fothergill and Furness, 2018*; *Grunddal et al., 2016*; *Habib et al., 2012*; *Sykaras et al., 2014*), and these hormone cocktails are not released directly into blood but rather into the lamina propria in the vicinity of the axons of vagal and spinal afferents (*Latorre et al., 2016*; *Najjar et al., 2020*). Some EECs likely form direct synaptic connections with these afferent terminals (*Bellono et al., 2017*; *Kaelberer et al., 2018*), and different hormones can be packaged into distinct vesicle pools within the same EEC (*Fothergill et al., 2017*; *Fothergill and Furness, 2018*). This spatial and functional organization of EECs creates potential mechanisms for the same hormone to have different effects depending on where it is released.

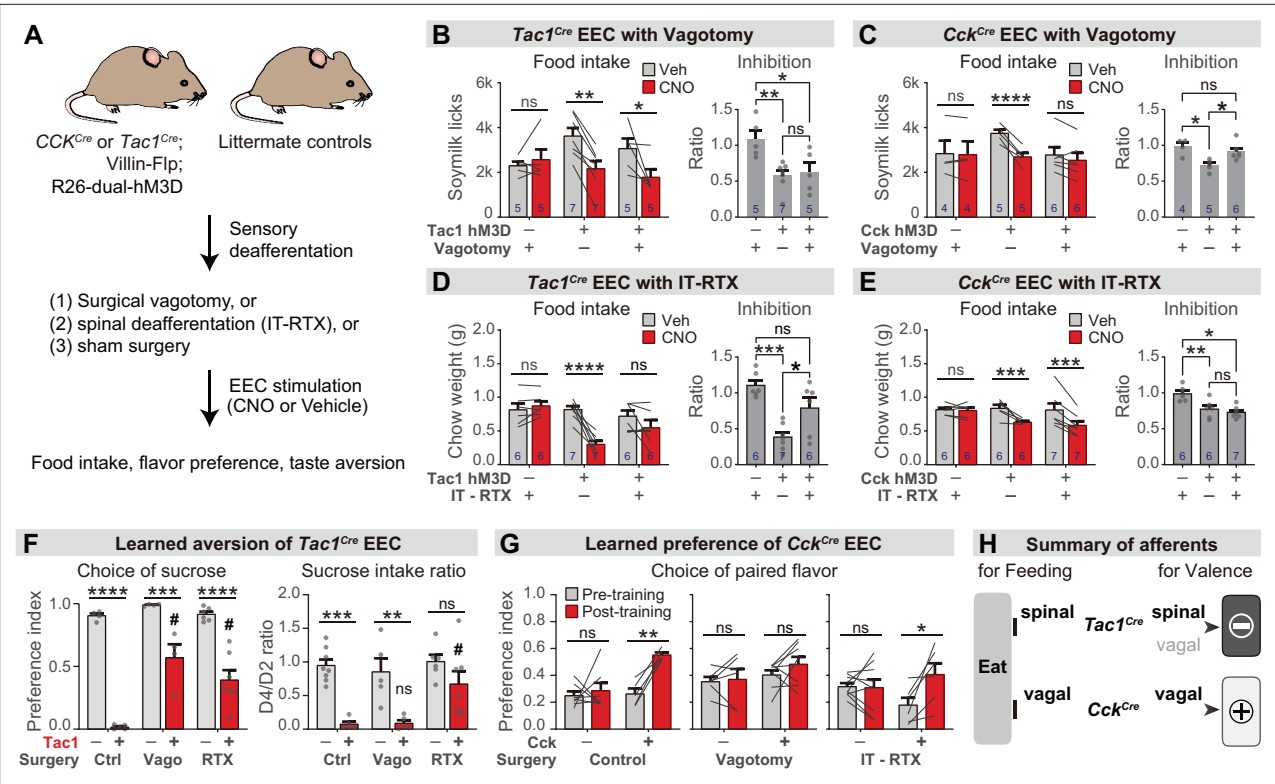

**Figure 6.** Parallel neuronal pathways convey EEC signals to the brain. (**A**) Experimental design. Triple-transgenic mice expressing hM3D in CCK or TAC1 EECs, or littermate controls, were subjected to either surgical vagotomy to remove subdiaphragmatic vagal afferents, or treatment with intrathecal RTX to ablate spinal afferents, or sham surgery. They were then treated with CNO or vehicle and the effect on feeding behavior and learned preferences was measured. (**B,D**) $Tac1^{Cre}$; $Vil$-$Flp$; $R26^{Dual\text{-}hM3D}$ mice and littermate controls were subjected to either subdiaphragmatic vagotomy or sham surgery (**B**) or IT-RTX or sham surgery (**D**). Animals were then food deprived overnight, injected with CNO or vehicle, and one-hour food intake was recorded. Right: ratio of feeding inhibition after treatment with CNO versus vehicle. Statistics for (B, left): Two-way repeated measures ANOVA. Drug: $F_{(1, 14)}=15.03$, p=0.0017. Subject: $F_{(2, 14)}=0.7452$, p=0.4925. Interaction: $F_{(2, 14)}=6.480$, p=0.0102. (B, right): Ordinary one-way ANOVA. $F_{(2, 14)}=6.926$, p=0.0081.(D, left): Two-way repeated measures ANOVA. Drug: $F_{(1, 16)}=20.25$, p=0.0004. Subject: $F_{(2, 16)}=4.673$, p=0.0252. Interaction: $F_{(2, 16)}=13.38$, P=0.0004. (D, right): Ordinary one-way ANOVA. $F_{(2, 16)}=13.40$, p=0.0004. (**C,E**) $Cck^{Cre}$; $Vil$-$Flp$; $R26^{Dual\text{-}hM3D}$ mice and littermate controls were subjected to either subdiaphragmatic vagotomy or sham surgery (**C**) or IT-RTX or sham surgery (**E**). Animals were then food deprived overnight, injected with CNO or vehicle, and one-hour food intake was recorded. Right: ratio of feeding inhibition after treatment with CNO versus vehicle. Statistics for (C, left): Two-way repeated measures ANOVA. Drug: $F_{(1, 12)}=32.50$, p<0.0001. Subject: $F_{(2, 12)}=0.6638$, p=0.5328. Interaction: $F_{(2, 12)}=15.33$, p=0.0005. (C, right): Ordinary one-way ANOVA. $F_{(2, 12)}=7.267$, p=0.0086. (E, left): Two-way repeated measures ANOVA. Drug: $F_{(1, 16)}=34.54$, p<0.0001. Subject: $F_{(2, 16)}=1.036$, p=0.3774. Interaction: $F_{(2, 16)}=7.123$, p=0.0061. (E, right): Ordinary one-way ANOVA. $F_{(2, 16)}=11.03$, p=0.0010. (**F**) $Tac1^{Cre}$; $Vil$-$Flp$; $R26^{Dual\text{-}hM3D}$ mice (red) and littermate controls (gray) underwent sham treatment, vagotomy, or IT-RTX before CTA training as in **Figure 4**. All mice received CNO during CTA training. Left: Preference for sucrose in the two-bottle test after CTA training. Ordinary two-way ANOVA. Genotype: $F_{(1, 24)}=154.1$, p<0.0001. Treatment: $F_{(2, 24)}=12.83$, p=0.0002. Interaction: $F_{(2, 24)}=7.792$, p=0.0025. Right: Ratio of total sucrose intake in the second sucrose training session (day 3, after pairing with CNO for one trial) versus first sucrose session (day 1, naive mice). Ordinary two-way ANOVA. Genotype: $F_{(1, 30)}=35.61$, p<0.0001. Treatment: $F_{(2, 30)}=4.673$, p=0.0171. Interaction: $F_{(2, 30)}=2.398$, p=0.1081. Of note, the apparent increased preference for sucrose after vagotomy is largely due to the low ingestion of water; see also **Figure 6—figure supplement 1I**. (**G**) $Cck^{Cre}$; $Vil$-$Flp$; $R26^{Dual\text{-}hM3D}$ mice and littermate controls underwent sham treatment, vagotomy, or IT-RTX and then were subjected to CFP training as in **Figure 4**. Shown is the preference for the CNO paired flavor in the two-bottle test before training (gray bars, day 1) and after training (red bars, day 6). Control: Two-way repeated measure ANOVA. Genotype: $F_{(1, 10)}=16.95$, p=0.0021. Time: $F_{(1, 10)}=9.746$, p=0.0108. Interaction: $F_{(1, 10)}=5.771$, p=0.0372.Vagotomy: Two-way repeated measure ANOVA. Genotype: $F_{(1, 12)}=1.866$, p=0.1970. Time: $F_{(1, 12)}=1.213$, p=0.2923. Interaction: $F_{(1, 12)}=0.4892$, p=0.4976. IT-RTX: Two-way repeated measure ANOVA. Genotype: $F_{(1, 13)}=0.08803$, p=0.7714. Time: $F_{(1, 13)}=5.795$, 0.0316. Interaction: $F_{(1, 13)}=6.671$, p=0.0227. (**H**) Summary of neuronal pathways required for Tac1$^{Cre}$ EECs or Cck$^{Cre}$ EECs to regulate food intake and drive food learning. Values are reported as mean ± SEM. *p<0.05, **p<0.01, ***p<0.001, ****p<0.0001; # or ns above bars indicate p-value comparing with the vehicle control treatment of the same genotype (#$P$<0.05); two-way ANOVA (left of B-E); **F**, **G**), or ordinary one-way ANOVA (right of B-E). See also **Figure 6—figure supplement 1**, **Figure 6—source data 1**, **Source data 1**, **Figure 6—source data 2**, **Figure 6—source data 3**, **Figure 6—source data 4**, **Figure 6—source data 5**, .

The online version of this article includes the following source data and figure supplement(s) for figure 6:

**Source data 1.** Raw data for **Figure 6B**.

*Figure 6 continued on next page*

*Figure 6 continued*

**Source data 2.** Raw data for *Figure 6C*.

**Source data 3.** Raw data for *Figure 6D*.

**Source data 4.** Raw data for *Figure 6E*.

**Source data 5.** Raw data for *Figure 6F*.

**Source data 6.** Raw data for *Figure 6G*.

**Figure supplement 1.** Parallel neuronal pathways convey EEC signals to the brain.

**Figure supplement 1—source data 1.** Raw data.

We investigated this by directly stimulating EEC subtypes in vivo and measuring the effect on behavior, which revealed many instances of divergence between EEC activation and hormone pharmacology. For example, we found that direct stimulation of *Cck^Cre* cells both inhibits feeding and produces robust conditioned flavor preference (*Figure 4I*), despite the fact that systemic injection of many of the individual hormones produced by these cells induces aversion at doses that inhibit food intake (*Chelikani et al., 2006*; *Deutsch and Hardy, 1977*; *Ervin et al., 1995a*; *Ervin et al., 1984*; *Halatchev and Cone, 2005*; *Kanoski et al., 2012*). We also found that the same hormone produced by different EECs can have different functions. For example, a PYY receptor antagonist blocked the anorexia produced by stimulation of *Cck^Cre* cells but not *Tac1^Cre* cells, even though PYY was expressed by subsets of both EECs (*Figure 5A and B*). Conversely, a 5-HT3R antagonist blocked the anorectic effect of stimulating *Tac1^Cre* but not *Cck^Cre* EECs (*Figure 5A and B*), although many *Cck^Cre* EECs also produce 5-HT (*Figure 3I*). These observations are consistent with the idea that EEC signaling is highly organized within the lamina propria, such that neurotransmitters released by EEC subtypes are funneled to specific sensory pathways. The ability to selectively access EEC subtypes for genetic manipulations will enable further studies that map these visceral circuits.

### Gut-brain pathways for food reward and aversion

Post-ingestive signals shape our decisions about what to eat and can override innate preferences, such that animals learn to prefer bitter foods that are nutritious and avoid sweet substances that are toxic (*Kern et al., 1993*; *Lin et al., 2017*; *Myers and Sclafani, 2006*). The gut epithelium is the interface between ingested food and the body and is poised to provide feedback about ingested food that drives learning. However, it has been unclear to what extent there are dedicated sensors in the epithelium that encode the values of ingested substances and drive food preference versus aversion. We therefore investigated the role of EEC subtypes in this process of learning about food.

The major finding from these experiments was that stimulation of two different EEC types was sufficient to drive either food preference or aversion, and that the effects of these cells were mediated by parallel gut-brain pathways that involve distinct hormones and afferent nerves. Acquisition of food preference was driven by stimulation of EECs labeled by *Cck^Cre* or *Gcg^Cre* and was blocked by antagonists against CCKAR but not any other hormone produced by these cells. Of note, prior work has shown that CCK injections can induce flavor preference at low doses that do not affect feeding (*Pérez and Sclafani, 1991*) and aversion at higher doses (*Deutsch and Hardy, 1977*; *Ervin et al., 1995a*; *Ervin et al., 1995b*). Our data show that direct stimulation of *Cck^Cre* cells can simultaneously produce both satiety and food reward.

The effect of *Cck^Cre* cells on satiety and food preference required intact vagal afferents, but not spinal afferents, which is consistent with an extensive literature showing that the vagus nerve mediates many of CCK's effects (*Joyner et al., 1993*; *Raybould, 2007*; *Ritter and Ladenheim, 1985*; *Smith et al., 1981*; *Smith et al., 1985*). The CCKAR is expressed in multiple vagal subtypes, including IGLE mechanoreceptors that innervate the stomach (*Glp1r+*) and intestine (*Oxtr+*) (*Bai et al., 2019*; *Williams et al., 2016*). These IGLE subtypes inhibit food intake when stimulated (*Bai et al., 2019*) and are required for the reduction in feeding following CCK injection (*Borgmann et al., 2021*) and therefore likely contribute to the effects of *Cck^Cre* EECs on food intake. It is less clear which vagal cells mediate CCK's effects on flavor preference, although stimulation of all gut-innervating vagal afferents is reinforcing (*Han et al., 2018*). One plausible candidate is the *Vip+/Uts2b+*vagal neurons that innervate the intestinal mucosa and express high levels of CCKAR but do not inhibit food intake when stimulated (*Bai et al., 2019*).

Peripheral 5-HT has been implicated in numerous processes including GI motility, secretion, vaso-constriction, bone growth, inflammation, visceral malaise, metabolism, and appetite (*Mawe and Hoffman, 2013*; *Spohn and Mawe, 2017*). Enterochromaffin cells are the major source of 5-HT in the gut, and we found that direct stimulation of *Tac1*<sup>Cre</sup>-labeled enterochromaffin cells acutely inhibited food intake and caused dramatic CTA. The inhibition of food intake required 5-HT signaling, whereas the CTA could be fully prevented only by simultaneously blocking receptors for two hormones produced by these cells during training (5-HT3R and TACR1). This is consistent with the hypothesis that some biological functions of EECs are encoded in their ability to release cocktails of hormones with redundant functions.

Intestinal serotonin can activate both vagal and spinal pathways, but we found that only spinal afferents were required for the reduction in food intake and CTA triggered by stimulation of enterochromaffin cells. This may be mediated by 5-HT3R-expressing spinal afferents that innervate the distal intestine (*Hockley et al., 2019*), are functionally coupled to enterochromaffin cells (*Bellono et al., 2017*), and are poised to transmit aversive visceral signals (such as abdominal pain) that are sensitive to 5-HT3R antagonists (*Ford et al., 2009*; *Gebhart and Bielefeldt, 2016*). Of note, many vagal sensory neurons also express 5-HT3Rs, and our findings do not rule out a role for those cells in nausea (*Babic and Browning, 2014*) or non-aversive satiety (*Savastano and Covasa, 2007*). However, we did not observe either aversion or 5-HT3R-sensitive anorexia following stimulation of *Cck*<sup>Cre</sup> EECs (which act via the vagus and also produce 5-HT). The ability to manipulate EEC subtypes in vivo should enable further studies that investigate how these pathways coordinate diverse physiology using a small set of neurochemical signals.

## Materials and methods
### Animals
Animals were maintained in temperature- and humidity-controlled facilities with 12 hr light-dark cycle and ad libitum access to water and standard chow (PicoLab 5053). We used the following transgenic mice, all of which were on a C57Bl/6 J background: *Cck*<sup>Cre</sup> (JAX 012706), *Gcg*<sup>Cre</sup> (JAX 030542), *Tac1*<sup>Cre</sup> (JAX 021877), *Sst*<sup>Cre</sup> (JAX 028864), *Ghrl*<sup>Cre</sup> (JAX 029260), *Fev-Cre* (JAX 012712), *Neurog3-Cre* (JAX 006333), *Slc17a6*<sup>Cre</sup> (JAX 016963), *Slc17a7*<sup>Cre</sup> (JAX –23527), *Slc17a8*<sup>Cre</sup> (JAX 028534), *R26*<sup>LSL-tdTomato</sup> (JAX 007914), *R26*<sup>Dual-hM3Dq</sup> (JAX 026942), *Igs7*<sup>tm162.1(tetO-GCaMP6s,CAG-tTA2)Hze</sup> (JAX 031562), and *Vil-Flp* (this study). Mice were at least 6 weeks old at the time of surgery. All studies employed a mixture of male and female mice and no differences between sexes were observed. All experimental protocols were approved by the University of California, San Francisco IACUC following the National Institutes of Health guidelines for the Care and Use of Laboratory Animals.

### Generation of Villin-Flp transgenic mice
The *Villin-Flp* allele was generated by pronuclear microinjection of *Villin-Flp-WPRE-pA* transgene. Briefly, the 12.4kb-Villin promoter was acquired from Addgene (12.4 kb Villin-ΔATG, Plasmid #19358). This promoter has been demonstrated to drive expression specifically in the intestinal epithelium and not in other tissues (*1*). After Xmal/XhoI linearization, a *Flpo-WPRE-hGH polyA-EcoR1* fragment was inserted after the *Villin* promoter using Gibson Cloning to generate *Villin-Flp-WPRE-polyA* flanked by EcoR1.

The construct was linearized by EcoR1 and prepared for pronuclear microinjection at the UCSF Gladstone Transgenic Gene-Targeting Core Laboratory. 120 injected embryos were implanted into pseudopregnant CD1 female mice. Seven out of 31 pups contained the Flp cassette and could transmit it to the next generation. Those seven founders were crossed to Flp-dependent reporter *RC::FLTG* (Jackson 026932) to characterize the pattern of Flp-mediated recombination. Two out of seven founders induced efficient recombination in the intestinal epithelium with very sparse labeling of the pancreatic cells (~1%), and one of them was kept as the *Villin-Flp* line and used for this study. The other five founder lines were eliminated, including three founders with good intestinal labeling as well as high-recombination in the pancreas (~90%), and two founders with poor recombination in the intestine. The *Villin-Flp* line was backcrossed to C57BL/6 J for more than six generations and was maintained on C57BL/6 J background.

## Generation of EEC scRNA-seq dataset (lineage-trace strategy)

Four 6-week-old *Neurog3-Cre; R26*<sup>LSL-tdTomato</sup> mice ( two males and two virgin females) were used for the scRNA-Seq preparation. Mice were anesthetized under isoflurane and then euthanized via cervical dislocation. Small intestines were harvested and cleaned in HBSS. Crypts and villi were isolated by 1 mM EDTA in HBSS (without $Ca^{2+}$, $Mg^{2+}$) for 10 min, mechanically detached, and pelleted. The crypt and villus pellets were then resuspended in warm digestion solution (2.5% Trypsin in HBSS) and incubated for 20 min, pelleted, washed with MEM, and triturated with a P1000 pipette. The suspended cells were filtered through a 40-μm cell strainer, pelleted again and resuspended in FACS buffer (0.1% BSA in PBS). TdTomato+ cells were collected using flow cytometry (yield 0.6–0.8% of total cells), and 198.5 k cells were suspended in 100 μl FACS buffer to reach a density of 2000 cell/ μL for sequencing.

Single cells were processed through the GemCode Single Cell Platform using the GemCode Gel Bead, Chip and Library Kits (10 X Genomics, Pleasanton) as per the manufacturer's protocol. Single-cell cDNA libraries were sequenced on the Illumina Highseq 4,000. De-multiplexing alignment to the mm10 transcriptome and unique molecular identifier (UMI)-collapsing were performed using Cell Ranger version 2.0.1, available from 10 X Genomics with default parameters. A gene-barcode matrix was generated. 0.3 billion reads over 7701 cells were captured, with 39.5 thousand mean reads per cell and 1.8 thousand median genes per cell.

## Integrated analysis of EEC scRNA-Seq datasets

To pre-process the EEC scRNA-seq datasets, the feature-barcode matrices were loaded into Seurat (v3) following standard workflow. For the lineage-trace dataset (generated in this paper), cells that fulfilled the following criteria (7001 cells covering 18,173 genes) were included for analysis: (1) percentage of mitochondrial reads <10%; (2) number of genes > 1000. For the pulse-trace dataset (*2*), cells that fulfilled the following criteria (2405 cells covering 20,751 genes) were included for analysis: (1) percentage of mitochondrial reads <10%; (2) number of genes > 250 and<9000; (3) number of reads >2000 and< 100,000.

The filtered scRNA-seq datasets (lineage-trace and pulse-trace) were further integrated using Seurat's IntegrateData function (anchors = 2000, dimensions = 1:40). Non-EEC epithelial cell types were identified based on published marker genes (*3*, *4*), including progenitors (*Neurog3*<sup>+</sup>), enterocytes (*Fabp1*<sup>+</sup>, *Reg1*<sup>+</sup>), goblet cells (*Guca2a*<sup>+</sup>, *Muc2*<sup>+</sup>), Paneth cells (*Lyz1*<sup>+</sup>, *Defa17*<sup>+</sup>), and tuft cells (*Dclk1*<sup>+</sup>, *Trpm5*<sup>+</sup>) (*Figure 1—figure supplement 1F*).

## Bulk RNA-Seq of CCK+ and CCK- EEC

*Cck*<sup>Cre</sup>; *R26*<sup>LSL-tdTomato</sup> mice (6–8 weeks old; overnight fasted or fed) were anesthetized under isoflurane and then euthanized via cervical dislocation. Small intestines were harvested and cleaned in HBSS. Crypts and villi were isolated by 1 mM EDTA in HBSS (without $Ca^{2+}$, $Mg^{2+}$) for 10 min, mechanically detached, and pelleted. The crypt and villus pellets were then resuspended in warm digestion solution (2.5% Trypsin in HBSS) and incubated for 20 min, pelleted, washed with MEM, and triturated with a P1000 pipette. The suspended cells were filtered through a 40-μm cell strainer, pelleted again and resuspended in FACS buffer (0.1% BSA in PBS). TdTomato+ cells and tdTomato- cells were collected (in FACS buffer) separately using flow cytometry (each independent dataset represents the TdTomato+ or TdTomato- cells from one mouse). RNA was extracted immediately after FACS and purified using the RNAeasy Micro kit (QIAGEN). RNA sample quality was checked using RNA PicoChip on a bioanalyzer (Agilent RNA 6000 Pico Kit). Amplified cDNA was prepared using Ovation RNA-Seq System V2, and the sequencing library was prepared using the Ovation Ultralow DR Multiplex system and sequenced on an Illumina Hiseq 2,500 platform.

Raw fastq reads were trimmed by trim_galore (v0.6.7) to remove residual sequencing adapter sequences and low-quality reads. Reads were mapped to the *Mus musculus* reference transcriptome (GRCm39.cdna.all.release-105) and quantified by Salmon (v1.6.0). The Bioconductor package `tximeta`(1.12.4) was used to convert transcript-level abundance to gene-level counts. Genes that have no counts or only a single count across all samples were removed. `DESeq2`(v1.34.0) was used to perform count normalization and differential expression analysis (with false discovery rate threshold alpha set to 0.05).

## Perfusion and tissue preparation

Mice were anesthetized under isoflurane and then transcardially perfused with 10 ml PBS followed by 15 ml formalin (10%). Brain, vertebral column, and visceral organs were dissected, post-fixed in 10% formalin overnight at 4 °C, and washed 3 × 20 min with PBS at RT. Tissues were kept in PBS at 4 °C before imaging, sectioning, or staining. For imaging myenteric ganglia (*Figure 3—figure supplement 1*), the duodenum was dissected, pinned flat in 10% formalin and stored overnight at 4 °C. After being rinsed 2 × 15 min in PBS at RT, the muscle layer was dissected away from the submucosa using a dissecting scope and sharp forceps. The muscle layer containing the myenteric ganglia was stored free-floating in PBS at 4 °C until immunohistochemistry for 2 hr at RT.

## Immunohistochemistry

Tissues were cryoprotected with 30% sucrose in PBS overnight at 4 °C, embedded in OCT, frozen and stored at –20 °C. Sections (30 um for peripheral tissues, or 50 um for the brain) were prepared with a cryostat and collected in PBS or on Superfrost Plus slides. Sections were washed 3×10 min with 0.1% PBST (0.1% Triton X-100 in PBS), blocked (5% NGS or NDS in 0.1% PBST) for 30 min at RT, and incubated with primary antibodies (diluted in blocking solution) overnight at 4 °C. For HuC/D staining of myenteric ganglia, primary antibody was incubated for 72 hr at 4 °C. The next day, sections were washed 3 × 10 min with 0.1% PBST, incubated with secondary antibodies (1:500 diluted in blocking solution) for 2 hr at RT, washed again 3×10 min with 0.1% PBST, and mounted using fluoromount-G with DAPI (Southern Biotech).

Primary antibodies used were: chicken anti-GFP (Abcam, ab13970, 1:1000), rabbit anti-GFP 1:1000 (LifeTech, A11122), goat anti-mCherry (ACRIS, AB0040-200), Rabbit anti-5-HT 1:2000 (Immunostar, 20080), Rabbit anti-CCK 1:500 (Millipore/Sigma,C2581), Rabbit anti-ChgA 1:1,000 (Immunostar, 20085), Rabbit anti GIP 1:500 (Abcam, ab22624), Mouse anti Glp1 1:200 (Abcam, ab26278), Rabbit anti PYY 1:500 (Abcam, ab22663), Rat anti-SST 1:200 (Millipore, MAB354), Rat anti-Substance P (TAC1) 1:200 (Abcam, ab7340), Guinea pig anti-TRPV1 1:500 (Millipore, AB5566), and rabbit anti-HuC/D 1:500 (Abcam, ab210554).

## Image acquisition

All histology images were taken by confocal microscopy (Zeiss, LSM 510) as previously described (*Bai et al., 2019*).

## Subdiaphragmatic vagotomy

Mice were anaesthetized with ketamine/xylazine delivered intraperitoneally (100 mg/kg Ketamine with 10 mg/kg Xylazine). A 1–2 cm incision was made along the medial line beginning at the distal edge of the sternum. The liver was then retracted with sterile cotton swabs that had been moistened with saline so that the distal end of the esophagus could be visualized. Both branches of the vagus nerve were isolated from the esophagus and a 1–2 mm section of each branch of the nerve was resected with scissors. Control mice for vagotomy experiments underwent a sham surgery that included internal organ manipulation but not vagotomy. To maintain gastrointestinal flow after vagotomy surgery, mice were kept on a mixed liquid diet (Enfamil, ASIN B004L5L5TA) and solid diet (standard chow, PicoLab 5053). Mice were allowed to rest for 2–3 weeks before used for behavior experiments.

To validate the subdiaphragmatic vagotomy, mice received an intraperitoneal injection of wheat germ agglutinin conjugated to Alexa Fluor 488 (WGA-488, 5 mg/kg, dissolved in PBS) and were euthanized 4 days later. WGA-488 is taken up by axon terminals of intact vagal motor neurons, the somas of which are located in the dorsal motor nucleus of the vagus (DMV) of brainstem and can be visualized by histology (*Figure 6—figure supplement 1A, B*). Labeling in the dorsal motor nucleus of the vagus was greatly reduced by subdiaphragmatic vagotomy.

## IT-RTX injection

For ablation of TRPV1$^+$ DRG neurons, 6- to 8-week-old mice were injected intrathecally with RTX (50 ng/mouse). Briefly, mice were lightly held and a 30 G needle attached with Hamilton syringe was inserted at the L6-S1 vertebral junction. For the RTX group, 10 µL of RTX (5 ng/uL in ethanol) was injected to reach a dose of 50 ng/mouse. For the vehicle control group, 10 µL of ethanol was injected instead. Mice were allowed to rest for 3 weeks before being used for behavior experiments.

Successful treatment was confirmed by a loss of TRPV1$^+$ neurons in the DRG, as well as reduction of TRPV1$^+$ signals of the superficial lamina of spinal cord dorsal horn (*Figure 6—figure supplement 1C, D*).

### Post-surgical care

Post-surgery mice were placed over a heating pad and monitored for their recovery from anesthesia. The health of the mice was monitored daily post-surgery. All mice were given post-surgery analgesia by subcutaneous injection of meloxicam (5 mg/kg) on post-surgery day 2 and day 3.

### Behavioral equipment

All experiments were performed in behavioral chambers (Coulbourn Instruments, Habitest Modular System). Feeding experiments were performed using a pellet dispensing system (Coulbourn, H14-01M-SP04 and H14-23M) with free water access. Food pellets (20 mg Bio-Serv F0163) were dispensed at the beginning of trials, or after pellet removal with a 10-s interval. Consumption of water or other liquid solution was monitored with contact lickometers (*Slotnick, 2009*).

### Feeding behavior

Mice were habituated for one night to the chambers, water bottle, food pellets, and pellet dispensing systems before the first experiments. Prior to the test, mice were fasted (for fast-refeeding tests) overnight (15–20 hr), except *Figure 4D* in which mice were ad libitum fed (to examine whether activation of EECs can increase food intake). All tests were performed during the light cycle.

To measure food intake during chemogenetic activation of EEC subtypes, triple-transgenic mice *Marker gene-Cre; Villin-Flp; R26$^{Dual-hM3D}$* mice or littermate control mice (mice from the same cross that lacked Cre or Flp, and therefore do not express hM3D) were injected intraperitoneally with CNO (1 mg/kg in saline with 1% DMSO) or vehicle (saline with 1% DMSO) and immediately placed into behavior chambers with food pellets and water accessible. After each experiment, pellet consumption was measured by deducting the quantity of pellets left on the ground from the total food count. For vagotomy experiments, a liquid-diet (0.15 g/mL, Enfamil, ASIN B004L5L5TA) was used instead of food pellets to maintain gastrointestinal flow after vagotomy surgery, and in this case water was not provided.

### Conditioned taste aversion (CTA)

All mice were naïve to sucrose or CNO prior to the CTA. Mice were habituated for one night to the chambers and water bottle before the CTA experiment. CTA experiments were performed on 5 consecutive days in Coulbourn behavior chambers during the light cycle.

Prior to each day of CTA, mice were water deprived overnight (15–20 hr). On day 1 and day 3, mice had access to water for 20 min in behavior chamber, followed by an intraperitoneal injection of vehicle. After the injection, mice were placed back to the behavior chambers without water access for another 40 min, and then were returned to their home cage with water/food access for the rest of the day. On day 2, mice were given access to a novel 5% sucrose solution for 20 min in the same behavior chamber, followed by an injection of antagonist (only for *Figure 5C* and *Figure 5—figure supplement 1C-E*) and an injection of CNO or vehicle (for all CTA experiments). After the injection, mice were placed back to the behavior chambers without sucrose access for another 40 min, and then were returned to their home cage with water/food access for the rest of the day. This conditioning was repeated on day 4. On day 5, mice were given access to two bottles (water and 5% sucrose) for 30 min.

To access the conditioned taste aversion, we calculated the choice of sucrose over water during the two-bottle test on day 5, using the sucrose choice index:

$$\text{Choice index (sucrose)} = \text{sucrose licks/total licks}$$

The sucrose intake ratio was also calculated, comparing sucrose intake after one trial of conditioning (day 4) versus the naïve condition (day 2):

$$\text{Intake ratio (sucrose)} = \text{Sucrose licks}_{Day4}/\text{Sucrose licks}_{Day2}$$

## Conditioned flavor preference (CFP)

Prior to the CFP, mice were habituated for one night to the chambers and water bottle. In a separate day, mice were habituated for one night to the flavored solutions in their home cage (with access to regular food and water). All mice were naïve to CNO prior to the experiments. The CFP were performed on 6 consecutive days during the light cycle.

Prior to each day of the CFP experiment, mice were water deprived overnight (15 hr). On day 1, the baseline choice of flavored solution was examined using two-bottle test. Mice were given access to two flavored solution in the Coulbourn behavior chamber for 10 min. These two bottle tests were performed in four trials in the same day with about 1 hr between each trial. Positions of the two bottles were swapped in trials 2 & 4 versus trial 1 & 3 to eliminate the effect of bottle's position on animal's preference. The choice of flavor (A or B) was calculated using the total licks across four trials:

$$\text{Choice index } (\text{flavor B}) = \text{Total licks}_B / (\text{Total licks}_A + \text{Total licks}_B)$$

On day 2 to day 5, training (conditioning) was carried out by coupling one of the flavored solutions with either CNO or vehicle ingestion. Each day, mice were exposed to one of the flavored solutions containing either CNO or vehicle for 2 hr within their home cage (without food and water). After that, mice were kept in home cage without food or water for another 2 hr, before they gained free access to food and water for the rest of the day. On day 2 and day 4, mice were exposed to their preferred flavor (A) containing 0.05% DMSO, while on day 3 and day 5, mice were exposed to their less-preferred flavor (B) containing 0.2 mg/ml CNO and 0.05% DMSO.

On day 6, the choice of flavored solution was examined again using two-bottle test, following the same protocol as day 1. The conditioned flavor preference was accessed by the choice of flavor paired with CNO during conditioned training:

$$\text{Choice index } (\text{flavor B}) = \text{Total licks}_B / (\text{Total licks}_A + \text{Total licks}_B)$$

The following pairs of flavored solutions were used in CFP (*Tan et al., 2020*): cherry versus grape flavors, or lemon-lime versus orange flavors. These included: Kool-Aid Lemon-Lime (SKU 00043000955444), 0.5 g/L with 1 mM AceK; Kool-Aid Orange (SKU 00043000955314), 0.5 g/L with 1 mM AceK; Kool-Aid Cherry (SKU 00043000955628), 0.9 g/L with 2 mM AceK; Kool-Aid Grape (SKU 00043000955635), 0.4 g/L with 1 mM AceK.

## Pharmacology

To observe the effect of antagonists on the feeding response to EEC stimulation, *Marker gene-Cre; Villin-Flp; R26*[Dual-hM3D] or littermate control mice were injected intraperitoneally with antagonist/vehicle followed by CNO/vehicle. Immediately after two consecutive injections, mice were placed into behavior chambers with food pellets and water accessible. In between tests, mice were provided in their home cage with ad libitum access to the same pellets used during testing, in order to discourage the development of any learned associations between the pellets and the stimuli delivered during a test (*Chen et al., 2016*). The lack of learned changes in consumption was confirmed by performing a control trial (pellet consumption following vehicle injection) at the beginning and end of each series of tests for each receptor antagonist. Due to the large number of antagonist/CNO/vehicle combinations, the testing sequence was not fully counterbalanced. Most animals were tested first with devazepide and ondansetron and then with the remaining compounds.

To observe the effect of antagonists on conditioned taste aversion, antagonist or vehicle was injected intraperitoneally prior to CNO/vehicle injection immediately after 20 min sucrose exposure on day 2 and day 4. For the conditioned flavor preference, antagonist or vehicle was injected intraperitoneally before mice were exposed to the less-preferred flavor with CNO.

Doses were chosen based on previously published reports: devazepide (R&D Systems, 2304) 1 mg/kg in saline with 1% DMSO, 5% Tween 80; ondansetron (Sigma, O3639) 1 mg/kg in saline, JNJ-31020028 (MedChemExpress, HY-14450) 10 mg/kg in saline with 5% DMSO and 5% Tween 80; RP67580 (Tocris, 1635) 1.5 mg/kg in saline with 1% DMSO; Exendin-3 (ApexBio, B6943) 10 μg/mouse in saline.

## Behavioral analysis

Statistical analyses and bar graphs of compiled behavioral data were generated using Prism. For feeding tests, two trials of the same treatment for each mouse were combined, averaged, and treated as a single replicate. All chemogenetic trials involved age-matched littermates as controls where possible.

To calculate pellet food intake, consumption of each pellet was defined as the first pellet removal event after each food pellet delivery. The food dropping ratio ($R_{drop}$) was calculated by the pellets found dropped divided by the total number of pellets removed at the end of each trial:

$$R_{drop} = Pellet_{drop}/Pellet_{removal}$$

At each time point, food intake was estimated by scaling the removed pellet with $R_{drop}$ :

$$Food\ intake = (1-R_{drop})$$

## EEC quantification

To investigate the identity of EECs labeled by each Cre lines, 30 mm sections of small intestine were prepared and are stained for the reporter and a panel of EEC markers (5-HT, SubP, PYY, CCK, GLP-1, GIP, and SST). The numbers of cell that were labeled by reporter, EEC markers, or both were quantified manually under confocal microscopy (Zeiss, LSM 510), and the overlap between reporter and EEC markers were quantified for villi or crypts separately. The density of cells was reported as number per distance of intestine section (*Figure 3—figure supplement 1B*, bottom):

$$Density\ of\ EEC = number\ of\ EEC/length\ of\ section$$

## Statistical analysis

All statistical analyses were performed using GraphPad Prism9. All values are reported as mean ± SEM (error bars or shaded area). Sample size is the number of animal subjects per group and is annotated within Figures or legend. p Values for comparisons across multiple groups were performed using analysis of variance (ANOVA) and corrected for multiple comparisons using Sidak multiple comparisons test. In Figures, asterisks denote statistical significance, $*p<0.05$, $**p<0.01$, $***p<0.001$, $****p<0.0001$. # above bars indicate statistical significance comparing with the vehicle control treatment of the same genotype, $\#p<0.05$.

# Acknowledgements

We thank the UCSF Genomics Core Facility and UCSF Gladstone Transgenic Gene-Targeting Core Laboratory for facility support, and João M Braz from Allan Basbaum's lab for assistance with the intrathecal injection. We thank Y Chen, C Zimmerman, E Feinberg, J Garrison, and members of the Knight lab for comments on the manuscript, and Julia Kuhl for illustrations.

# Additional information

### Funding

| Funder | Grant reference number | Author |
| --- | --- | --- |
| National Institutes of Health | R01-DK106399 | Zachary A Knight |
| National Institutes of Health | RF1-NS116626 | Zachary A Knight |
| Howard Hughes Medical Institute | Investigator | Zachary A Knight |
| Eli Lilly and Company | | Zachary A Knight |

| Funder | Grant reference number | Author |
|--------|------------------------|--------|

The funders had no role in study design, data collection and interpretation, or the decision to submit the work for publication.

## Author contributions

Ling Bai, Conceptualization, Data curation, Formal analysis, Investigation, Visualization, Methodology, Writing – original draft, Writing – review and editing; Nilla Sivakumar, Sheyda Mesgarzadeh, Tom Ding, Truong Ly, Timothy V Corpuz, James CR Grove, Brooke C Jarvie, Investigation; Shenliang Yu, Data curation, Formal analysis, Visualization; Zachary A Knight, Conceptualization, Supervision, Funding acquisition, Writing – original draft, Project administration, Writing – review and editing

## Author ORCIDs
Sheyda Mesgarzadeh ⬦ http://orcid.org/0000-0003-0138-5566
Zachary A Knight ⬦ http://orcid.org/0000-0001-7621-1478

## Ethics

All experimental protocols were approved by the University of California, San Francisco IACUC (protocol #AN179674) following the National Institutes of Health guidelines for the Care and Use of Laboratory Animals.

## Decision letter and Author response
Decision letter https://doi.org/10.7554/eLife.74964.sa1
Author response https://doi.org/10.7554/eLife.74964.sa2

# Additional files

## Supplementary files
• Source data 1. Summary statistics.
• Transparent reporting form

## Data availability

Source data is included in the manuscript. RNA-seq data is available from the Gene Expression Omnibus (GSE203200). Villin-Flp mice have been deposited at Jackson laboratory.

The following dataset was generated:

| Author(s) | Year | Dataset title | Dataset URL | Database and Identifier |
|-----------|------|---------------|-------------|--------------------------|
| Bai L, Knight Z, Yu S | 2022 | Enteroendocrine cell types that drive food reward and aversion | https://www.ncbi.nlm.nih.gov/geo/query/acc.cgi?&acc=GSE203200 | NCBI Gene Expression Omnibus, GSE203200 |

The following previously published dataset was used:

| Author(s) | Year | Dataset title | Dataset URL | Database and Identifier |
|-----------|------|---------------|-------------|--------------------------|
| Gehart H, van Es JH, Hamer K, Beumer J, Kretzschmar K, Dekkers JF, Rios A, Clevers H | 2018 | Identification of functional enteroendocrine regulators by real-time single-cell differentiation mapping | https://www.ncbi.nlm.nih.gov/geo/query/acc.cgi?acc=GSE113561 | NCBI Gene Expression Omnibus, GSE113561 |

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
