## [Editor Report]

This study provides insight into the functional diversity of specialized cells in the gut using a combination of transcriptomics, genetics, and behavioral assessment. It demonstrates how select enteroendocrine cell types mediate food reward while others drive aversion.

---

## [Decision Letter]

**Decision letter after peer review:**

Thank you for submitting your article "Enteroendocrine cell types that drive food reward and aversion" for consideration by *eLife*. Your article has been reviewed by 3 peer reviewers, including Alexander Chesler as the Reviewing Editor and Reviewer #1, and the evaluation has been overseen by Kate Wassum as the Senior Editor.

The reviewers have discussed their reviews with one another, and we have drafted this to help you prepare a revised submission.

Essential revisions:

As you will see below, the reviews are overall positive, and the reviewers agreed this is a useful dataset and important technical approach to understand and study EEC function. While no new experiments are requested, we are asking for significant edits. We'd like to see the data presented in a manner that allows the reader to identify and understand the key findings. For example, representative images should better match the quantified data and some information would seem better suited as supplemental figures; we would also like more informative figure legends. Questions were also raised the interpretation of some of findings and worry that some of the descriptions are overstated. For example, since food intake in a hungry mouse might be inhibited by many things including visceral distress or by a feeling of satiety, it is perhaps less surprising than the manuscript suggests that activation of different EEC populations have distinct effects on associative learning. Similarly, are the appetitive effects described really robust? We would like better clarification on the methods. For example, given the conditioning effects of EEC activation, the methods should provide details about the order in which mice were treated with drugs in the pharmacological survey, and whether treatment order was counterbalanced. One reviewer asked that the references be expanded to include previous work in the field, where appropriate. Finally, questions were raised about the variability of the control data that seem greater than other studies so we would like these at least commented on.

The complete reviews are included below. Please provide point by point responses to both the comments above and each reviewer when you submit your revised manuscript.

*Reviewer #1 (Recommendations for the authors):*

Some points that if address would enhance the manuscript:

1) While I'm mostly positive about the work, I found the data presentation confusing and at times almost impenetrable. While transparency in data reporting is welcomed, the way it is presented here made it very difficult to identify and evaluate the key findings.

2) Additionally, the large variation across controls is somewhat surprising, given feeding assays like these in hungry mice tend to be very robust and reproducible across strains. In some cases, the difference across strains seems larger than the manipulations.

3) The figures are full of control information that could (and should) be in supplementary figures. For the most part the authors are comparing Tac1 vs CCK-positive EECs and the main figures would benefit from focusing on these two groups of cells.

4) Lastly, the example images in Figure 2H seem to have little correlation with the numbers quantified data in 2I. Furthermore, in 2I, the data are plotted differently in the right and left histograms.

*Reviewer #3 (Recommendations for the authors):*

Some points to consider for making the manuscript more informative to the reader:

1. Figure 2: it is very difficult to understand exactly what is being plotted in (i) and how it relates to the data shown in (h) and the supplement e.g., Tac1-Cre vs SubP.

2. Figure 3 shows food intake varies widely in vehicle treated mice and has very different dynamics in the various experimental animals suggesting mechanisms for suppression of intake might be different for different drivers. Similarly, in Figure 4, the effects of CNO also vary between groups making the effects of antagonists potentially misleading. The authors may wish to reassess or at least should explain some of these aspects. As another example of the sorts of problems a reader faces, Figure 4a is entitled: Tac1Cre EEC induced feeding inhibition requires 5-HT3R, but this does not seem an accurate reflection of the data or to fit with the title of Figure 4 figure supplement 1a: Tac1Cre EEC induced feeding inhibition requires TACR1 and NPY2R, which also doesn't fit what the figure seems to show.

3. The data in Figure 4 and Figure 4 figure supplement 1 on conditioned taste aversion are at first sight compelling but are also quite confusing in that preference index and intake ratio often do not match up. The authors comment that vagotomy decreases water intake (and show this in Figure 5 figure supplement 1), but it is unclear how that would fit with other lick data shown after vagotomy that don't seem to change. Moreover, there are also discrepancies between the preference index and the intake ratio when inhibitors are used. Therefore, it is difficult to assess if for example RP reverses all CTA as would seem from intake ratio or if ODS and RP have additive effects as suggested by preference index. Clearly, the conclusions drawn are affected by the choice of assay that is considered most relevant.

---

## [Author Response]

Essential revisions:As you will see below, the reviews are overall positive, and the reviewers agreed this is a useful dataset and important technical approach to understand and study EEC function. While no new experiments are requested, we are asking for significant edits. We'd like to see the data presented in a manner that allows the reader to identify and understand the key findings. For example, representative images should better match the quantified data and some information would seem better suited as supplemental figures; we would also like more informative figure legends.

We agree that some of the data could have been presented more clearly, and we thank the reviewers for their suggestions. We have made several changes to improve the data presentation.

1) We have extensively edited the figure legends to make the descriptions clearer and more detailed, especially for Figures 5 and 6. The figures should now be interpretable without reference to the text.

2) We added white arrows to indicate all the co-localized cells in Figure 3H, so that the reader can easily evaluate how the representative images in 3H correlate with the quantified data in 3I.

3) We have modified Figure 3I (quantification of co-localization between markers and Cre drivers) in several ways to simplify the interpretation.

First, we inserted an example interpretation directly above each graph: (left) “fraction PYY+ that are also *Fev^Cre^* +” and (right) “fraction *Fev^Cre^* + that are also PYY+”.

We also included a detailed explanation in the figure legend.

“Quantification of co-localization between different EEC subtypes and the cells labelled by various *Marker gene-Cre; RosaLSL-Reporter* mice (example images are shown in Figure 3H). The bar graphs on the left show the fraction of EEC subtypes (Marker+) that are also labelled by each Cre line. For example, the fraction of cells that stained for PYY that are also labelled by *Fev^Cre^* recombination. The bar graphs on the right show the fraction of the cells labelled by each Cre line that also stained for the marker gene. For example, the fraction of cells labelled by *Fev^Cre^* recombination that also stained for PYY. Comparing the left and right reveals, for example, that *Fev^Cre^* labels a high percentage of cells for all EEC subtypes, but that each EEC subtype represents only a subset of the *Fev^Cre^* labelled cells. This quantification was performed separately for cells located in the intestinal villi (dark gray bars) and crypts (light gray bars).”

We have also changed the layout of Figure 3I so that the right and left sets of bar graphs have the same layout.

(4) We have adjusted the coloring schemes for Figures 5 and 6 to better highlight the data from the experimental cohorts (bright red) and deemphasize the control groups (gray).

(5) We have modified Figure 5D to make it clear that the comparison is before and after training.

(6) We have simplified the schematics in Figure 3A, 4A, and 6A.

Questions were also raised the interpretation of some of findings and worry that some of the descriptions are overstated. For example, since food intake in a hungry mouse might be inhibited by many things including visceral distress or by a feeling of satiety, it is perhaps less surprising than the manuscript suggests that activation of different EEC populations have distinct effects on associative learning.

We agree that many things can inhibit food intake. A central point of our paper is that different EEC subtypes inhibit food intake in different ways (e.g. via positive or negative valence effects), and that these differences are reflected in assays of learning.

Similarly, are the appetitive effects described really robust?

Yes. The conditioned flavor preference induced by CCK stimulation shown in Figure 4 is reproduced with different cohorts of mice in Figure 5 and Figure 6.

We would like better clarification on the methods. For example, given the conditioning effects of EEC activation, the methods should provide details about the order in which mice were treated with drugs in the pharmacological survey, and whether treatment order was counterbalanced.

We have expanded the methods to provide more detailed information. For all experiments involving food intake measurements, mice were provided, in between tests, with ad libitum access in their home cage to the same test pellets used during testing. We have previously shown that this can prevent mice from developing conditioned associations with the test pellets that promote or inhibit consumption (Chen, Lin et al. 2016). We confirmed this here by measuring food intake in the absence of CNO/antagonist at multiple timepoints before and after testing each set of drugs. For experiments involving conditioning (CFP and CTA), we used naïve mice, and those mice were not used again in any subsequent experiments.

One reviewer asked that the references be expanded to include previous work in the field, where appropriate.

We have included the references related to vagotomy requested by the reviewer. We have also included a discussion, references, and additional data regarding the expression of vesicular glutamate transporters in EECs, as requested by the Reviewer 2. We have incorporated these into new Figure 2 (discussed below).

Finally, questions were raised about the variability of the control data that seem greater than other studies so we would like these at least commented on.

We have included a discussion of this in the detailed responses to points raised by the reviewers. Briefly, we do not believe the control data is unusually variable, and we control for any variability by including both genotype controls (mutant versus littermate) and within-animal comparisons (e.g. drug vs. vehicle; before and after training). The latter provide an internal control for variability in baseline food intake.

Reviewer #1 (Recommendations for the authors):Some points that if address would enhance the manuscript:1) While I'm mostly positive about the work, I found the data presentation confusing and at times almost impenetrable. While transparency in data reporting is welcomed, the way it is presented here made it very difficult to identify and evaluate the key findings.

We have modified the figures and legends to make the interpretation simpler. These changes are detailed in lines 9-38 above.

2) Additionally, the large variation across controls is somewhat surprising, given feeding assays like these in hungry mice tend to be very robust and reproducible across strains. In some cases, the difference across strains seems larger than the manipulations.

Different assays have different levels of variability, but the measurements of simple food intake by hungry mice in our study are not highly variable. Plotted in Author response image 1 is the food intake from the vehicle treated animals in Figure 4c.

**Author response image 1. sa2fig1:** 

In Figure 4F, there is some variability in how much sucrose different cohorts drank during CTA training. However, the effect we report for *Tac1^Cre^* is all-or-none, and this effect is replicated using independent cohorts of animals in Figure 5 and 6.

Measurements of conditioned flavor preference (in Figures 4, 5 and 6) are inherently variable compared to simple food intake measurements, because each animal starts with an idiosyncratic preference and the overall effect sizes are smaller. However, we control for these sources of variation by including genetic controls (co-housed littermates +/- Cre/Flp) as well as within-animal controls (drug vs. saline; before and after training) in almost every experiment. The within-animal comparisons control for baseline differences. Moreover, we replicate the key flavor preference results from Figure 4 with independent cohorts of mice in Figures 5 and 6.

Finally, it is important to note that vagotomy (Figure 6) causes well-known changes in baseline food intake, and these changes can be variable between animals. For this reason, the analysis of these experiments focuses on within animal comparisons (+/- drug or before/after training).

3) The figures are full of control information that could (and should) be in supplementary figures.

We appreciate the sentiment that some of the figures can seem complex due to the many controls. For the reasons stated above, we believe the control data is critical for evaluating the claims in the paper, and this control data is easier to evaluate if it is situated side-by-side with the experimental data on the same axes (as opposed to moved to the supplement).

To make the figures easier to interpret, we have extensively expanded the figure legends to highlight the relevant comparisons and spell out in detail the interpretation. For example, in the revised legend for Figure 5a,b we state explicitly which bars the reader should focus on for the key comparison. We have also made subtle changes to the layout so that the experimental groups are clearly highlighted and the control groups are deemphasized.

For the most part the authors are comparing Tac1 vs CCK-positive EECs and the main figures would benefit from focusing on these two groups of cells.

We agree that the comparison between TAC1 and CCK is the focus of the functional experiments in the paper. Figures 5 and 6 present data only from TAC1 and CCK cells. Figures 1 presents global RNA-seq data and so by necessity include data from all cell types. Figure 2 contains RNA-seq data that is focused primarily on CCK cells. Figures 3 and 4 describe the development and validation of the method and the initial screen of cell types, and so by necessity include data from the other cell types.

4) Lastly, the example images in Figure 2H seem to have little correlation with the numbers quantified data in 2I.

We agree that Figure 3H,I (formerly 2H,I) was confusing. We have addressed this by inserting white arrows to identify all the cells that overlap in the two channels in Figure 3H. This shows that the images in 3H are generally consistent with the quantification in 3I.

Furthermore, in 2I, the data are plotted differently in the right and left histograms.

We have adjusted Figure 3I (formerly 2I) so that the bar graphs in the right and left are plotted in the same order. We have also made other changes to make this figure clearer (described above).

Reviewer #3 (Recommendations for the authors):Some points to consider for making the manuscript more informative to the reader:1. Figure 2: it is very difficult to understand exactly what is being plotted in (i) and how it relates to the data shown in (h) and the supplement e.g., Tac1-Cre vs SubP.

We agree that this was confusing, and we have altered these figures and the legend to make them clearer. Please see the response to the Essential revisions for a discussion.

2. Figure 3 shows food intake varies widely in vehicle treated mice and has very different dynamics in the various experimental animals suggesting mechanisms for suppression of intake might be different for different drivers.

We agree it is likely that the different cell types have different mechanisms for inhibiting feeding (positive versus negative valence), and this is an important point of the paper. For a discussion of variability in control mice, please see the response to Reviewer #1 (lines 122-140).

Similarly, in Figure 4, the effects of CNO also vary between groups making the effects of antagonists potentially misleading. The authors may wish to reassess or at least should explain some of these aspects.

Again, we feel that the responses to CNO in Figure 4a and 4b are reasonably consistent given the number of independent experiments that are shown. It is important to note that we do not use the exact same animals for every drug tested in Figure 4a and 4b (due to cohort attrition, the timing of experiments, etc…), which likely contributes to some baseline variability between subpanels. However, the key statistical comparisons are within-animal, and we use the same cohort of animals for all the tests within each subpanel.

As another example of the sorts of problems a reader faces, Figure 4a is entitled: Tac1Cre EEC induced feeding inhibition requires 5-HT3R, but this does not seem an accurate reflection of the data or to fit with the title of Figure 4 figure supplement 1a: Tac1Cre EEC induced feeding inhibition requires TACR1 and NPY2R, which also doesn't fit what the figure seems to show.

We thank the reviewer for catching this mistake. The title of Figure 5- Supplement 1a (formerly Figure 4- Supplement 1a) has now been corrected so that it matches Figure 5a.

Figure 5a (formerly 4a) and the supplement both show that a 5-H3TR antagonist significantly reduces the inhibition of feeding caused by Tac1 stimulation. Based on this, we state in the figure title that 5-HT3R is required for this effect. The figures also show that the effect of the 5-HT3R antagonist may be enhanced by combination with a TACR1 antagonist, although the TACR1 antagonist on its own has no effect. We did not incorporate these additional observations into the figure title because they cannot be concisely described.

3. The data in Figure 4 and Figure 4 figure supplement 1 on conditioned taste aversion are at first sight compelling but are also quite confusing in that preference index and intake ratio often do not match up.

The major difference between in Figure 5c (formerly 4c) and the supplement 5e is that RP alone partially blocks the effect on choice but completely blocks the effect on intake during training. However, it is clear from examining Supplemental Figure 5e that there is high variability in the RP intake data during training (the data are essentially bimodal) and this variability is reduced in the Ods/RP combination. It is not uncommon for the choice and intake tests to give slightly different results, because the choice test is performed on animals that have completed two days of training, whereas the intake measurement is performed after one day of training and so animals have more variable levels of learning. The choice test is also intrinsically more sensitive to preference than intake, because the animals have a choice.

The authors comment that vagotomy decreases water intake (and show this in Figure 5 figure supplement 1), but it is unclear how that would fit with other lick data shown after vagotomy that don't seem to change.

Vagotomy causes gastroparesis that typically slows food and water consumption, and other lick data in the paper show that vagotomy reduces intake, although the effects are somewhat smaller; e.g., in Figure 6b,c (formerly 5b,c) compare the gray bars with and without vagotomy. In Figure 6, the combination of vagotomy and CTA training with Tac1 resulted in very low intake of both fluids in the final choice test. This may be due to a generalized aversion to the testing conditions caused by the combination of vagotomy and a repeated negative stimulus. Whatever the cause, the small number of licks in the choice test made the results difficult to interpret. For this reason, we emphasized the training data in this experiment.

Moreover, there are also discrepancies between the preference index and the intake ratio when inhibitors are used. Therefore, it is difficult to assess if for example RP reverses all CTA as would seem from intake ratio or if ODS and RP have additive effects as suggested by preference index. Clearly, the conclusions drawn are affected by the choice of assay that is considered most relevant.

Please see the comment above. We believe that if there are sufficient licks for a meaningful comparison, the choice test is the most relevant assay, because it is performed on animals that have completed the training and because it is more sensitive to preference. Regardless, we present all of this in the paper, including the raw data in the Source Data files, so that readers can evaluate the results for themselves.

References

Chen, Y., et al. (2016). "Hunger neurons drive feeding through a sustained, positive reinforcement signal." *ELife*
**5**.